

# Slope stability and rock fall hazard assessment of volcanic tuffs using RPAS and TLS with 2D FEM slope modelling

Ákos Török[1], Árpád Barsi[2], Gyula Bögöly[1], Tamás Lovas[2], Árpád Somogyi[2], and Péter Görög[1]

[1]Department of Engineering Geology and Geotechnics, Budapest University of Technology and Economics, Budapest, H-1111, Hungary
[2]Department of Photogrammetry and Geoinformatics, Budapest University of Technology and Economics, Budapest, H-1111, Hungary

*Correspondence to*: Ákos Török (torokakos@mail.bme.hu)

**Abstract.** Low strength rhyolite tuff forms steep cliffs in NE Hungary. A multi-dimensional approach including field analysis and laboratory tests was conducted to understand the mechanical properties of the tuff and to measure discontinuity surfaces. With the help of RPAS (Remotely Piloted Aircraft System) and TLS (Terrestrial Laser Scanning), a digital terrain model (DTM) was generated, and the results of these surveys were compared. Cross sections and joint system data were obtained from DTM and used as input parameters for the slope stability analyses. The rocky slope was modelled by 2D FEM (Finite Element Method) software and potential hazards such as planar failure, wedge failure and toppling were identified. The paper demonstrates the usefulness of combined field analyses, geomechanical laboratory testing and various remote sensing techniques (such as RPAS and TLS) in rock face stability calculations and failure mode analysis.

## 1 Introduction

Higher resolution geospatial data products have been developed at local, national, and global scale (Chen et al. 2016). Remotely piloted aerial systems can provide products that meet the requirements of national mapping (Cramer 2013), but also support applications such as coastal and glacial monitoring (Harwin and Lucieer 2012), monitoring of vegetation growth, military reconnaissance (Kostrzewa et al., 2003) and monitoring of forests (Rufino and Moccia, 2005; Scholtz et al., 2011; Fritz et al. 2013) and of forest fires (Rufino and Moccia 2005). These systems can be used in rapid mapping in emergency situations (Choi et al., 2009), in vegetation and/or biodiversity control but also enable the detection of several species such as orang-utans, elephants or rhinos and provide information on density and circulation of animals (Wich and Koh 2012), even counting birds (Grenzdörffer 2013). RPAS applications also cover the inspections of high and medium voltage lines, oil and gas pipelines, roads and railways (Colomina and Molina 2014), and can be used for data capturing of cultural heritage and archaeological sites (Rinaudo et al. 2012, Colomina and Molina 2014, Pajeres 2015).

In the past years, technological development of RPAS (Remotely Piloted Aircraft System) revolutionized the data gathering of landslide affected areas (Rau et al. 2011), recultivated mines (Haas et al. 2016) or road cuts (Mateos et al. 2016). These tools have been increasingly used in slope stability analyses (Niethammer et al. 2012) and in erosion process detection





(Neugrig et al. 2016). Another remote sensing method that has changed our understanding of landslide mechanism profoundly is terrestrial laser scanning (TLS) which is a useful tool in failure analysis by providing slope geometries (Fanti et al. 2012, Assali 2014, Franconi 2014). The drawback of the latter method is that it is impossible to apply from the ground at hardly accessible cliff faces. Thus, it is much better to obtain a reliable Digital Terrain Model (DTM) and slope geometries

with a combination of these tools. The application of these techniques brought significant amount of new data on previous landslides (Fanti et al. 2012, Neugrig et al. 2016) and also on failure forecast (Manconi & Giordan 2015). Rock falls represent special landslide hazards since their rapid movements and various trajectories make it difficult to predict their hazard potential (Costa & Agliardi 2003). Several methods have been suggested to assess cliff stability from physical prediction rock fall hazard index (Crosta & Agliardi 2003) via Rockfall Hazard Rating System (Budetta 2004) and to

modelling of their trajectories (Crosta & Agliardi 2002, Abbruzzese et al. 2009, Copons et al. 2009, Samodra et al. 2016). These methods rely on understanding failure mechanisms and on predicting displacement of rock masses (Pappalardo et al. 2014, Stead & Wolter 2015, Mateos et al. 2016) or at some cases individual rock blocks (Martino & Mazzanti 2014). To gather data on the rock fall hazard of existing cliff faces, a number of crucial data is needed: slope profiles, material properties, block size (De Biagi et al. 2017) and possible discontinuity surfaces that can contribute to slope instability. Slope

profiles can be obtained by TLS or RPAS, while material properties have to be measured on site (e.g. UCS by Schmidt hammer) or under laboratory conditions (Margottini et al. 2015). Detection and mapping of joints require fieldwork (on site measuring by compass), or at hardly accessible locations it is possible by applying remote sensing techniques (Fanti et al. 2012), or both.

Most of rock fall hazard publications deal with hard, well cemented rocks such as limestone (Samodra et al. 2016) or various

other types of sedimentary rocks (Michoud et al. 2012) igneous or metamorphic rocks. In contrast, very few previous studies deal with cliff face stability and rock fall hazard of low strength rock such as volcanic tuffs (Fanti et al. 2012, Margottini et al. 2015). Volcanic tuffs are very porous rocks and prone to weathering (Arikan et al. 2007). While the current paper deals with a low strength pyroclastic rock, it has a slightly different approach of cliff stability analysis, since slope stability is assessed by using a combination of remote sensing techniques, field measurements, and laboratory testing of tuffs with 2D

FEM (Finite Element Analysis) analyses of slopes. Compared to other case studies this study operates on a smaller scale and studies the possibilities of wedge and planar failures. The cliff face is unstable as it is evidenced by falling blocks. Due to rock fall hazard, the small touristic pathway was closed to avoid causalities. The current paper analyzes the cliff faces by condition assessment and stability calculations. Thus, this research provides an assessment of how the combination of TLS with RPAS can be used to create a surface model at hardly accessible sites. The paper also demonstrates the combined use of

photogrammetric, surveying, and engineering geological methods at difficult ground conditions in assessing rock slope stability.





## 2 Study area

The study area is located at mid mountain range in NE-Hungary. A hardly accessible jointed rhyolite tuff cliff face was studied. On the top of the cliff a touristic point, the Sirok Castle is located (Fig. 1). The steep rhyolite tuff hill with an elevation of 298 m AMSL and found at the transition area of two mountain ranges, Mátra and Bükk Mountains. The tuff is

5    very porous and prone to weathering (Török et al. 2007). It shows similarities to the volcanic tuffs of Capadocia (Aydan et al. 2003).

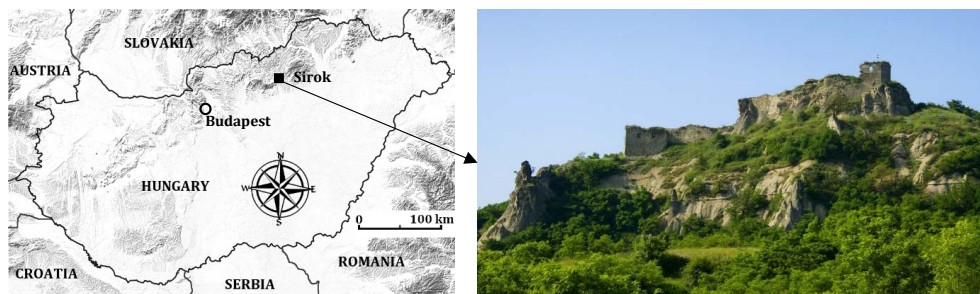

**Fig.1. Location of studied cliff faces and an image of the rocky slope at Sirok Castle, NE Hungary**

Although the first castle was already constructed in the 13th century AD, due to war damages and reconstructions, the current structure encompasses wall sections representing different construction periods. In these days, the partially ruined walls have been restored, and the castle is open to tourists but southern slopes are closed due to rock fall hazard.

The hill represents a rhyolite tuff that was formed during the Miocene volcanism (Badenian-Lower Pannonian period). It is a

15    part of the Inner Carpathian volcanic chain. The geological map of the closer area clearly reflects the dominance of pyroclastic rocks, with isolated occurrences of Triassic carbonates (Fig. 2). The cliffs are steep and display several joints and discontinuity surfaces (Fig. 3). The present study focuses on the southern hillslope of the castle hill, where major rock falls occurred in the near past (Fig. 4).




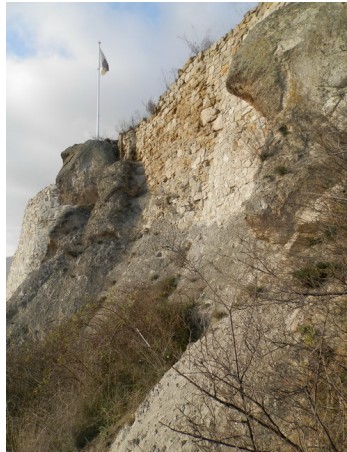

**Fig.2. Steep cliff faces at Sirok Castle hill, southern slope**

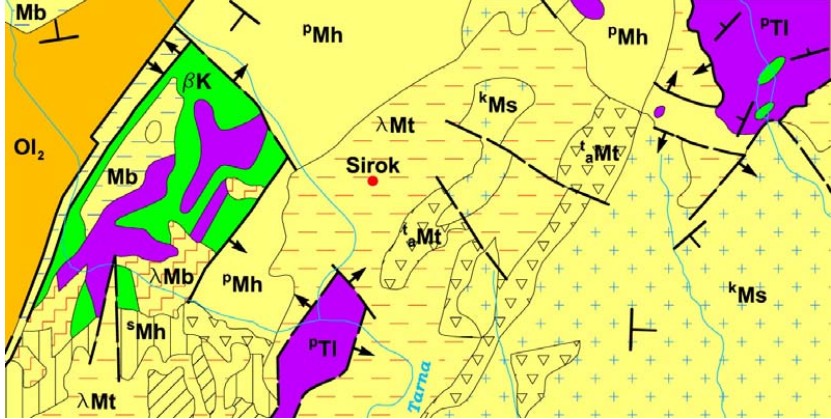

5     **Fig.3. Geological map of the area (redrawn after Balogh 1964) Legend: Tl- Triassic carbonates, K- Cretaceous volcanic rocks, Ol-Oligocene sediments M-Miocene pyroclastic rocks**


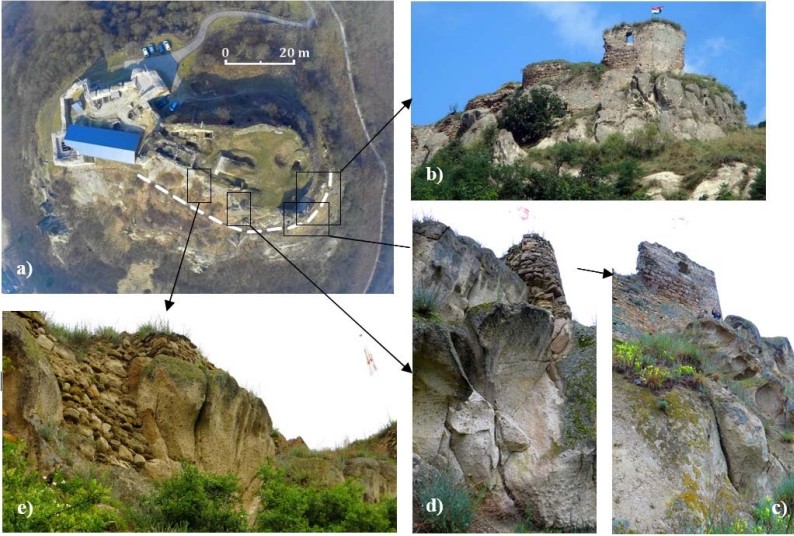

**Figure.4. Studied southern cliff faces (clockwise): a) image of the castle obtained by RPAS with marked details; b) distant view of the eastern part of the cliff section; c) weathered rounded cliff with larger taffoni; d) vertical to sub-vertical cliff face with steep joints and traces of rock fall; e) steep cliffs dissected by joints.**

## 3 Materials and Methods

### 3.1 Terrain data acquisition

Cliff stability analysis required the accurate 3D modelling of these highly dissected rock faces. Since major parts of the site consist of hardly accessible steep slopes that are partly covered by vegetation, traditional surveying was not possible. As a consequence, remote sensing technologies were applied. The first approach was to use only terrestrial laser scanning that allows rapid data acquisition and generation of 3D point clouds that – during post-processing – allows creating surface models. Due to slope geometries and limited access to the cliff faces, occlusions, and disadvantageous incident angles, the detection of all slope geometries was not possible by using only TLS. Hence, additional terrain data was obtained by using two types of RPAS of slopes that surround the southern part of the castle (Fig. 5).


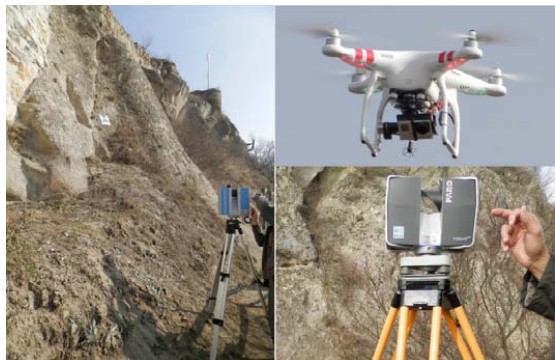

**Figure 5. Applied remote sensing techniques (clockwise): Z+F Imager; 5010C DJI Phantom 2 with the GoPro Hero3+ action camera and Faro Focus S 120 3D**

**3.2 TLS**

5 The terrestrial laser scanning (TLS) data were captured by two scanners, a Faro Focus S 120 3D (Faro, 2016) and a Z+F Imager 5010C (Z+F 2014). The terrestrial laser scanning was executed on 21st February 2015, when vegetation cover was limited and there was no snow cover. Although Faro scanner has less maximum measurement range than that of the Z+F (Table 1), it is a small, light scanner, easy to move and deploy, therefore this device was used on the top off the cliffs and on steep slopes. Both devices store all data on built-in memory cards.

**Table 1: Technical parameter of the applied terrestrial laser scanners**

| Scanner | Z+F Imager 5010C | Faro Focus S 3D 120 |
|---|---|---|
| **Numbers of scan stations** | 10 | 29 |
| **Resolution** | 3 mm/10 m | 3 mm/10 m |
| **Ranging accuracy** | 4 mm | 2 mm |
| **Maximal measurement range** | 187 m | 120 m |
| **Scanning frequency** | 1 016 000 points/s | 976 000 points/s |
| **Scanning wavelength** | 1500 nm | 905 nm |
| **Color information** | Yes | Yes |
| **Tie points** | Checkerboards | Spheres |
| **Web** | www.zf-laser.com | www.faro.com |





Noticeably, the scanners were used with the same resolution in order to obtain near equally dense point clouds. Similarly, both scanners have captured images to support identification during post-processing and to ensure providing enhanced quality end products (Spreafico et al. 2015).

Because of the steep slopes and cliffs, the scanners could acquire point clouds in a limited range, so the cloud matchings get higher priority. The tie objects were surveying markers, checkerboard plates of size A4, and spheres of diameter of 15 cm. During the entire time of the field work there were 38 tie points marked by spheres and 7 further tie points marked by checkerboards.

Due to georeferencing, particular tie objects had to be measured also by Global Navigation Satellite System (GNSS). The used GNSS receiver was a Leica CS10 with a Gs08plus antenna (GS08, 2014, CS10, 2014). The measurement was done in RTK mode supported by the Hungarian RTK network (RTKnet, 2013). There were 7 measured ground control points (GCPs); the mean 3D measurement accuracy was 4.9 cm (minimal value was 2 cm, maximal value 9 cm). The raw merged point cloud measured by Faro scanner contained 1.9 billion points, whilst the Z+F point cloud 0.8 billion points. After the resampling, these data sets were reduced to 110 million and 40 million points, respectively. Both point clouds have X, Y Z coordinates, intensity and RGB color values (FaroScene, 2012, CC, 2014 Matlab, 2007, LeicaCyclone, 2016). The point cloud processing chain included several steps (Fig. 6).

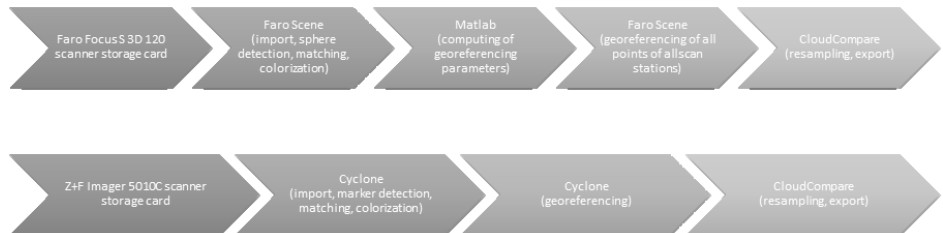

**Figure 6. Processing of the terrestrial laser scanning data and the applied software environments**

### 3.3 RPAS

The Remotely Piloted Aerial System (RPAS) (Eisenbeiss 2008) was deployed on the same day as the terrestrial laser scanning. The system is a modified commercial DJI Phantom 2 drone (DJI, 2016), where the flying vehicle has been equipped with a synchronous image transfer that also forwards the current flying parameters (e.g. height, speed, tilt, power reserve). For safety reasons, the crew consisted of two persons: one for controlling the aircraft, the other one for continuously monitoring the transferred video stream. The camera control is done by a tablet.





A GoPro Hero 3+ (GoPro, 2017) action camera was mounted onto a 2-DoF gimbal of the unmanned aerial vehicle (UAV). The camera has a fixed 2.77 mm focal length objective that is capable of capturing 4000 × 3000 pixel sized JPG images. The images were captured with a sensitivity of ISO 100 and sRGB color space. The lens was used with a fixed aperture of 2.8 and the camera was able to adjust the adequate shutter speed. Generally the exposure time was set to 1/1400 s and the images

were compressed at a rate of 4.5 bits/pixel. There were three imaging flights; two around noon and one about 5 in the afternoon. The flying times were 13, 12 and 13 minutes, respectively, where 390, 365 and 419 images were captured. All 1174 images were involved in the photogrammetric object reconstruction (Fig. 7). The photogrammetric reconstruction has been done by Pix4Dmapper (Pix4D, 2017), which is based on Structure-from-Motion (SfM) technology (Westoby et al. 2012, Danzi et al. 2013, Lowe 2004). After the image alignment, the image projection centres and attitudes can be observed

in (Fig. 8). 12 million points obtained by the photogrammetric reconstruction.

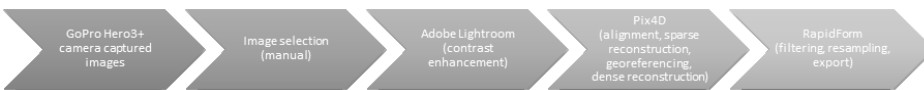

**Figure 7. Processing of the RPAS collected imagery**

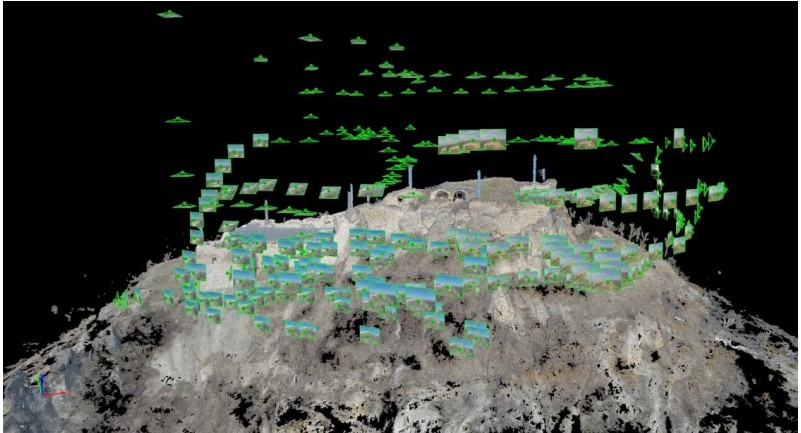

**Figure 8. The captured image positions around the reconstructed castle hill**



### 3.4 Processing of raw point clouds, creating DEM and spatial analyses

The considerable amount of points obtained is multiplications of the surface points. The redundancy is because of the doubled terrestrial laser scanning and of the UAV survey. Since the merged point cloud is difficult to be managed due to its size, and has heterogeneous point spacing, the later processing requires the elimination of the redundancy. First, each point

cloud was resampled by CloudCompare, where the spatial resolution of the point clouds was set to 1 cm. After this reduction, the laser scanned point clouds were merged and resampled to 1 cm spatial resolution.

The laser scanned point clouds were then imported into Geomagic Studio 2013 (GeomagicStudio, 2013) and the mesh model with 1 cm triangle side lengths was derived. The UAV point cloud was also imported and meshed, but the triangle side length was 5-7 cm.

To support the geological survey, several horizontal and vertical sections (Fig. 9) were derived in Geomagic DesignX 2016 (GeomagicDesignX, 2016); these profiles were exported in CAD format (DXF).

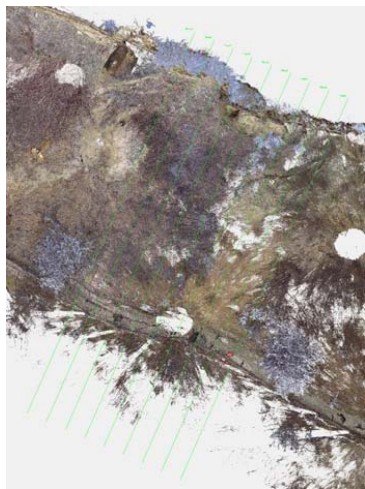

**Figure 9. Vertical sections marked by parallel green lines to support geologic analyses**

The next step was to make cut-offs focusing only on the cliffs; it was done by CloudCompare, followed by the TLS and UAV points being exported in LAS-format (LAS, 2012). The exported points could then be imported into SAGA GIS 2.1.2 (Conrad et al. 2015), where the necessary DEMs were created by inverse distance weighting (IDW) algorithm (IDW, 2013). The derived DEM-grids have 5 cm spatial resolution, which is adequate for morphologic analyses. The morphology analysis has concentrated on Catchment Area (CA), Stream Power Index (SPI) and Topographic Wetness Index (TWI) (Haas et al.

2016). These indices express the potential relationship between surface geometry and geological parameters.




The visualization is further supported by developing Streetview-like demonstration environment; a dedicated web-site is designed, where the users have an access of the surface image as well as the necessary navigation tools (move forward, backward, rotate and zoom). This solution has less importance in hazard evaluation but it gives the possibility to look around and check what is visible near the rocks.

**3.5 Engineering geological field measurements**

Field data collection formed the first part of engineering geological surveys. Major lithotypes were identified and described and geological profiles were recorded. Rock joints, discontinuity surfaces and fault systems were measured by using compass and structural geological software applied in mobile phone. The structural geological data was analyzed by Dips software. Strength parameters were assessed on site by using a Schmidt hammer. 10 rebound values were measured on each surface and mean values and standard deviations were also calculated. This method has been also used previously to gather rapid data on rock strength of cliff faces (Margottini et al. 2015). The data-set was compared to rock mechanical laboratory tests.

**3.6 Rock mechanical laboratory tests**

Samples for laboratory analyses were collected on site. Major rock mechanical parameters were measured under laboratory conditions on cylindrical specimens. These were drilled from blocks and cut into by appropriate size using cutting disc. The sizes of tested specimen were made according to EN on air dry and on water saturated samples. The specimens were grouped according to the bulk density and the propagation speed of the ultrasonic pulse wave. Strength parameters such as uniaxial compressive strength, an indirect tensile strength (Brasilian), was measured according to relevant EN standards and modulus of elasticity was also calculated (Table 2). The generalized Hoek-Brown failure criterion (Hoek et al. 2002) was used to determine strength parameters of the rock mass. Altogether, 53 cylindrical test specimens were used for the tests.

**Table 2. Rock mechanical tests and relevant standards.**

| Rock mechanical parameter | Number of specimens | Relevant standard |
|---|---|---|
| Bulk density | 53 | EN 1936:2000 |
| Water absorption | 18 | EN 13755:2008 |
| Propagation speed of the ultrasonic wave | 53 | EN 14579:2005 |
| Uniaxial compressive strength | 31 | EN 1926:2006 |
| Modulus of elasticity | 31 | ISRM 2015 |
| Tensile strength (Brasilian) | 23 | ISRM 2015 |





### 3.7. 2D FEM modelling

The stability analysis of the rocky slopes was focused on the southern slopes. The falling blocks can endanger the touristic footpath bellow the castle. The FEM modelling was made by the RocFall software of the Rocscience. The rock slope

stability of the investigated hillslope was analyzed according to Hudson & Harrison (1997). Since the rhyolite tuff is a weak rock with few joints the rock mass failure and the failure along discontinuities were also analyzed. Kasmer et al. (2013) used FEM software for stability analysis of Cappadocian tuffs. Shakmekhi & Tannant (2015) used FEM software for probabilistic rock slope stability analysis considering the variability of joint geometry.

First, the rock mass failure was analyzed with RS2 FEM software on the steepest sections which was determined according

to the TLS and UAV model. Muceku et al. (2016) show a complex slope stability analysis of a heritage town with the same FEM software.

Secondly, the kinematic analysis had been done with stereographical tool. The planes of the main discontinuity sets were measured manually on site. Many parts of the hillslope cannot have been measured manually, therefore, the TLS and UAV model had been used also to determine the most hazardous part of the hillslope for block stability analyses. Nowadays TLS

and UAV measurements are commonly used for rock slope stability analysis. Aliardi et al. (2013) used TLS measurements for investigation of structurally controlled instability mechanism. Tuckey & Stead (2016) investigated the determination of the two critical factors of rock slope instability: the persistence of discontinuities and intact rock bridges with remote sensing methods. Liu et al. (2017) used digital photogrammetry tools to determine the geometrical characteristic of discontinuities for block stability analysis. During the kinematic analysis planar sliding, wedge sliding, and toppling failure were tested with

Dips software. Probabilistic kinematic analysis had been done with grid based GIS method, as in Park et al. 2015, in order to map the dangerous zones of the rock slopes of a road. The determination of the safety factor of the dangerous blocks was made by LEM analysis with Rocscience software.

## 4. Results and discussion

### 4.1. Comparison of RPAS and TLS data

The two, basically different data collection techniques resulted in two point clouds covering the surface. These clouds were resampled, so their spatial resolution was homogenized. These two models are suitable to compare the data collection techniques (Fig. 10).





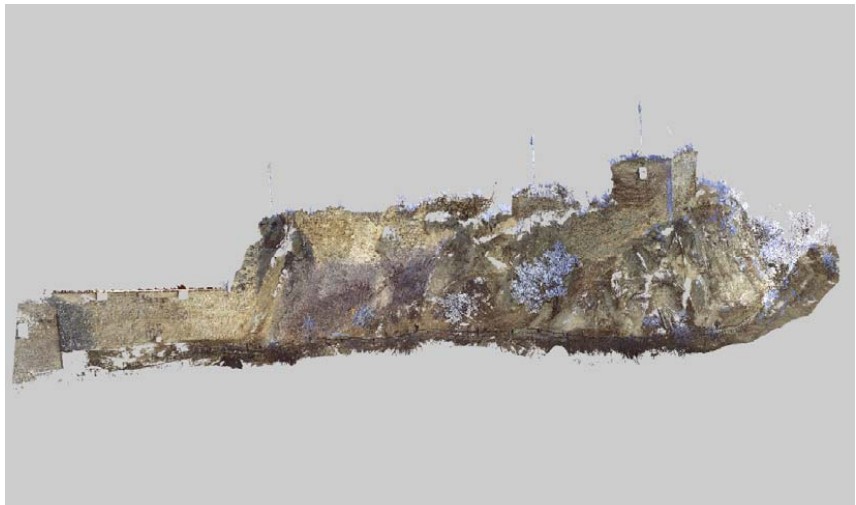

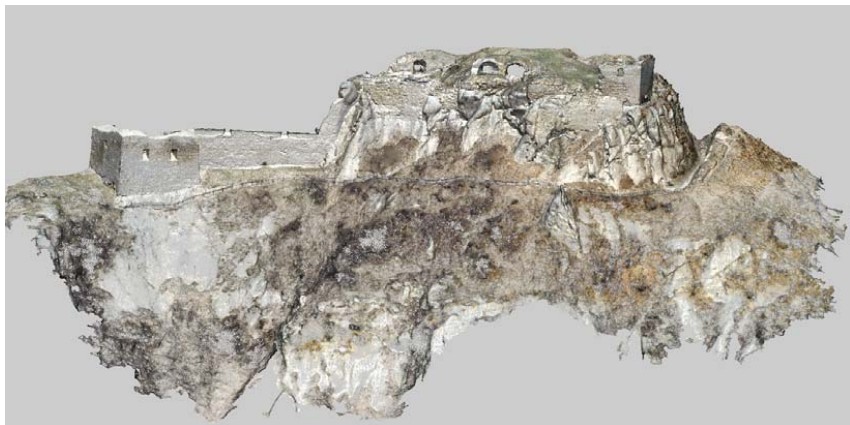

**Figure 10. The point clouds obtained by TLS (top) and by RPAS (bottom) technologies**

The raw point clouds were transferred into geographic information system, where the digital elevation models were created (Fig. 11).



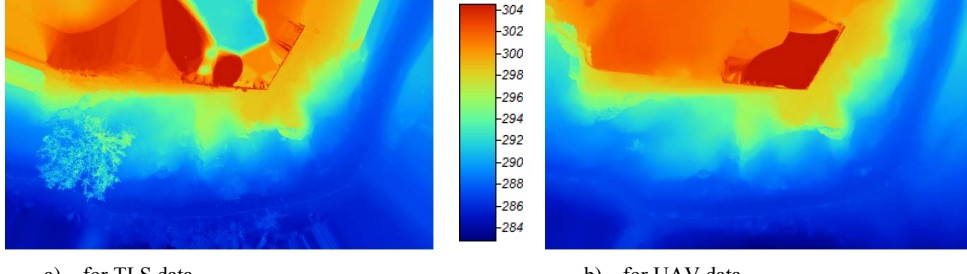

a) for TLS data          b) for UAV data

**Figure 11. The Digital Elevation Models**

First, the point density is tested; in CloudCompare, a unit sphere of volume of 1 m$^3$ was defined where the points can be counted and then the sphere can be moved along the whole surface. The point amounts in the unit spheres represent the data

5    collection density (Fig. 12. and Fig. 13).

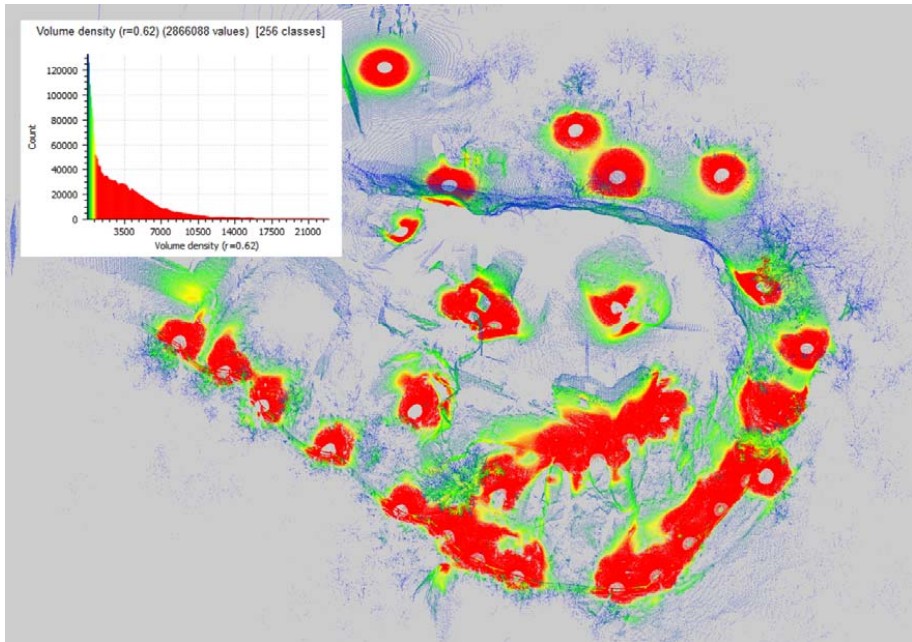

**Figure 12. TLS point cloud densities derived by the use of a 1 m$^3$ unit sphere**



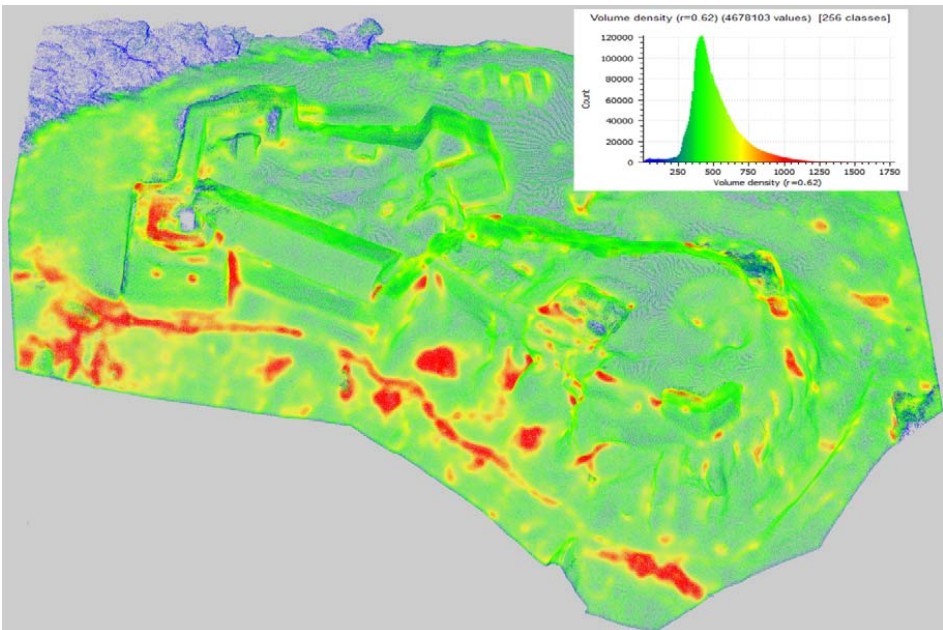

**Figure 13. UAV point cloud densities derived by the use of a 1 m³ unit sphere**

Obviously, the terrestrial laser scanning dataset has significantly higher point densities close to the scan stations, which is indicated in the figure by the red clusters (Fig. 12). The UAV data is much more homogeneous; it is because of the less original data density. Another aspect causing some differences between the two data sets is that the image based

5    reconstruction is performed by interest operators, very usually SIFT (Scale-invariant Feature Transform) or similar computer vision operators (Lowe 2004). These operators are generally sensitive to intensity jumps, points, or corners, and textural changes in the input images. If the image resolution is not adequate or the object is locally "smooth", these operators do not return with surface points and the output of the reconstruction has some "filtered" effect. This phenomenon is clearly visible on the cliffs; the smoothing effect can excellently be seen by the Analytical Hillshading (AH) (Fig. 14).





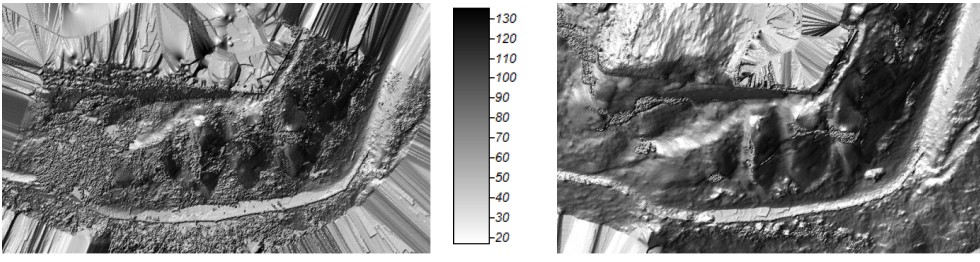

a) TLS data                                            b) UAV data

**Figure 14. Analytical hillshading as surface smoothness representation**

The digital elevation model can be analyzed in terrain morphological sense. The Catchment Area (CA), Stream Power Index (SPI) and the Topographic Wetness Index (TWI) derived for both data sets is shown in Fig. 15 to be compared to the UAV-image of the same area. All resulting morphological maps strongly express the already eroded and potentially unstable 'prone to erosion' parts of the cliff. Compared the two data sets, it is clearly proven that the geometric resolution of the UAV-based digital elevation model corresponds to the TLS one, so the faster, simpler, and cheaper UAV based surveying technology offers similar quality results as the terrestrial laser scanning (Fig. 14).





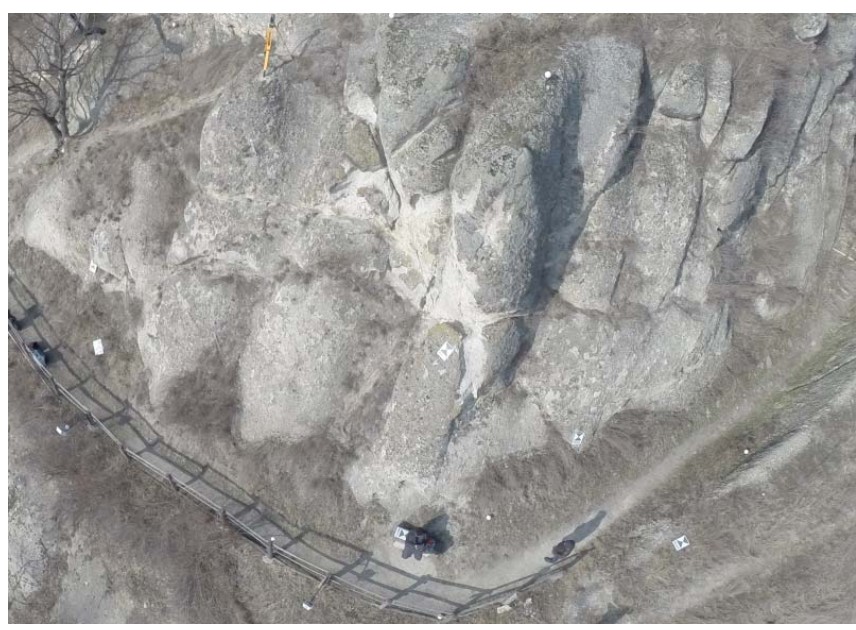

a)   UAV-captured image

b)   Catchment area for TLS DEM   c)   Catchment area for UAV DEM

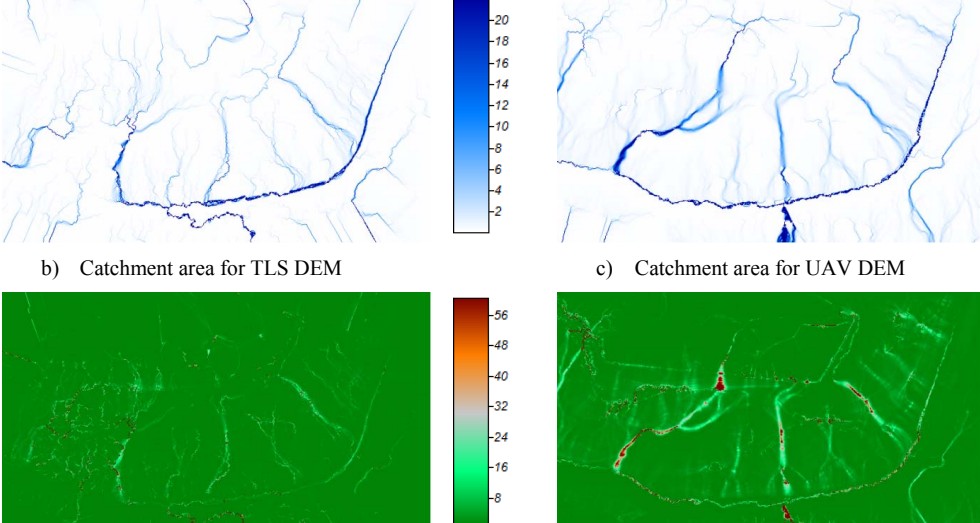

d)   Stream Power Index for TLS DEM   e)   Stream Power Index for UAV DEM



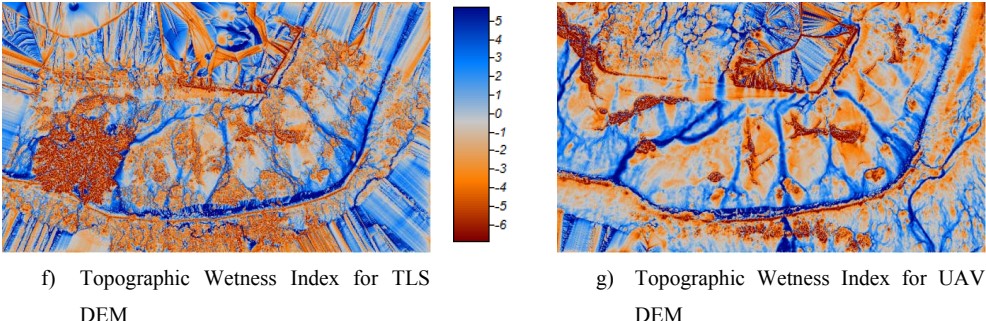

f) Topographic Wetness Index for TLS DEM

g) Topographic Wetness Index for UAV DEM

**Figure 15. Cliff and the corresponding morphological results for TLS and UAV data sets**

### 4.2. Compilation of engineering geological field and laboratory data

The rhyolite tuff faces consist of moderately bedded ignimbritic horizons and also brecciated lapilli tuffs and tuffs according

5  to our field observations (Fig. 16). The topmost 10 metres of the cliff face which was modelled from slope stability comprises 3 main horizons and can be modelled as "sandwich structure". The lower and the upper part are formed by thick pumice containing lapilli tuffs. These beds enclose nearly 2 metres of well-bedded less-welded fine tuff and brecciated horizons (Fig. 17).

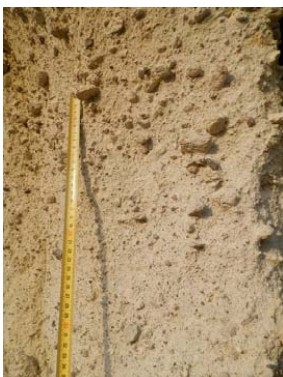

**Figure 16. Lapilli-tuff the dominant lithology of the studied cliff**


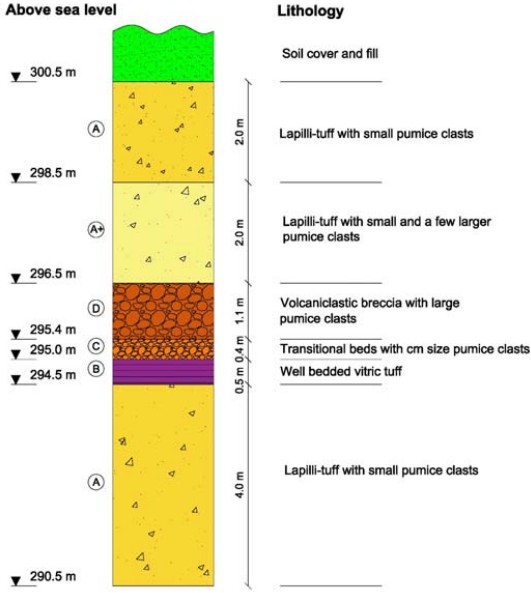

Figure 17. Lithologic column of Sirok Várhegy showing the modeled topmost 10 metres section of the hill

The orientation of discontinuities and joints was measured on the S and SE part of the hillslope. Discontinuities which are

5   close to the footpath were measured manually while the hardly accessible ones were identified on images obtained by TLS and UAV (Fig. 18).

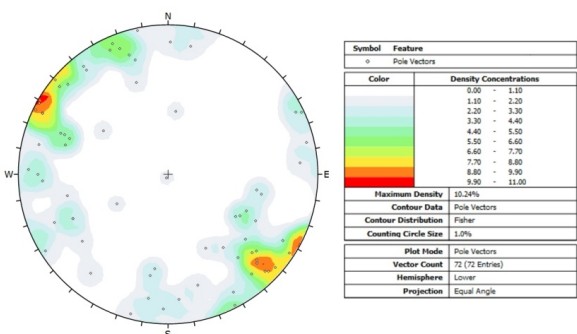

10   **Fig. 18. Stereographic plot of the discontinuities on the southern part of the hill**



Additional measurements were also made in the underground cellar system of the castle, where the tuff is also exposed. These identified discontinuities were also documented by TLS (Fig 19). Combining and comparing all measured data of discontinuities and joints a prevailing NE – SW on the southern hillside (Fig. 20).

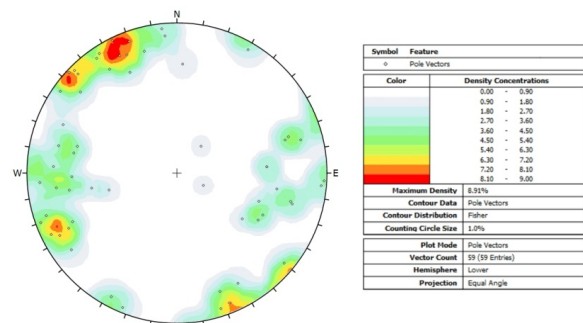

Figure 19. Stereographic plot of the discontinuities measured in the cellars

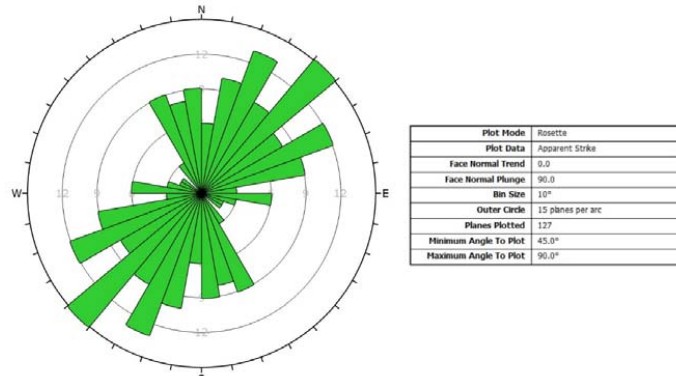

Figure 20. The frequency of joints measured on the castle hill

The laboratory tests of tuffs provided the input data for stability analysis for the two main lithologies: upper and lower unit of lapilli tuff and middle unit of less welded tuff (Table 3).

The GSI values were determined according to Marinos et al. (2005). In the model calculations GSI=50 value was used (Fig. 21).






Figure 21. GSI of moderately jointed rock mass of castle hill (GSI=50, see red circle) (table is from Marinos et al. 2005)

Table 3. Rock mechanical parameters of tuff used in the model: lapilli tuff refers to upper and lower 4 metres, less welded tuff refers to middle startigraphic unit

| Mechanical property | | Upper and Lower unit (Lapilli tuff) | Middle unit (Less welded tuff) |
|---|---|---|---|
| Bulk density($\rho$) | [kg/m$^3$] | 1815 | 1635 |
| Uniaxial compressive strength($\sigma_c$) | [MPa] | 8.02 | 0.35 |
| Tensile strength ($\sigma_t$) | [MPa] | 0.83 | 0.04 |
| Modulus of elasticity(E) | [GPa] | 0.97 | 0.05 |





### 4.4. Slope stability modelling

The global stability of the hillslope was calculated by using RS2 FEM software. The results suggest that the global factor of safety is SRF=1.27-1.71. The SRF factor is influenced by the weak tuff layer which has very low shear strength compared to the lapilli tuff. Our failure analyses have demonstrated that the bottom of the slip surface would be in the weaker layer (Fig.

5   22). These results indicate that failure occurs in the weak layer and could lead to a larger mass movement.

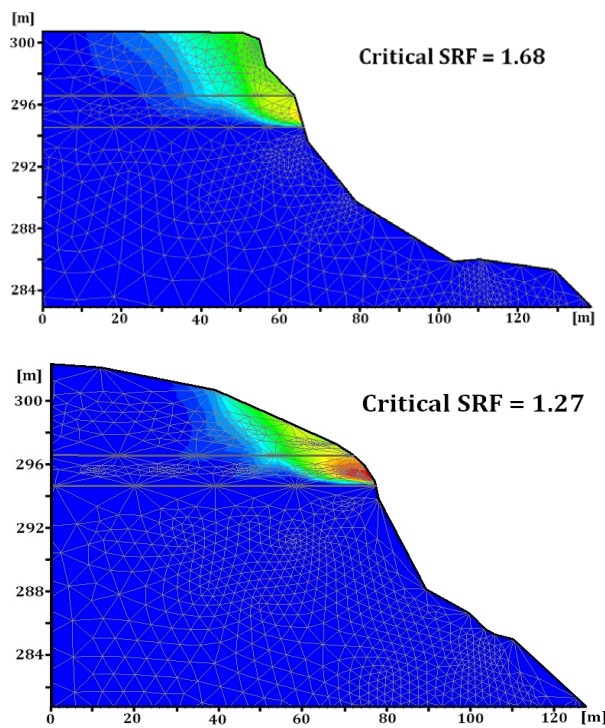

**Figure 22. The results of the global stability analysis of the slope, total displacements are marked in blue to red**

The Dips software was used for the kinematic analysis. The direction of the hillslopes and the direction of the discontinuities were compared to determine the location of the potential hazardous failure zones on the hillside. Possibility of planar sliding and wedge failure were analyzed. Toppling failure due to the geomorphology cannot occur. There was no regular spacing of the discontinuities. Stereographic plots were generated showing the possible failure planes for all slope directions. As an

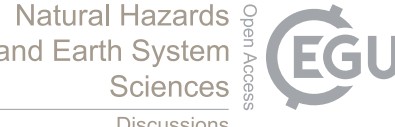
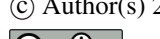

example the most hazardous part of the slope is shown here (75/75) (Fig. 23.). The safety factor of the possible sliding rock (planar failure) was calculated by Rocplane software (Fig. 24).

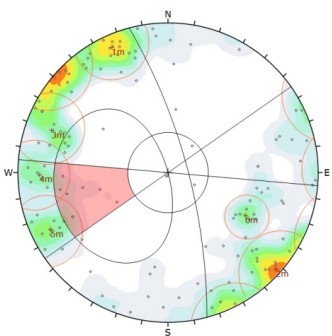

5    **Figure 23. Kinematic analysis of planar failure (slope: 75/75)**

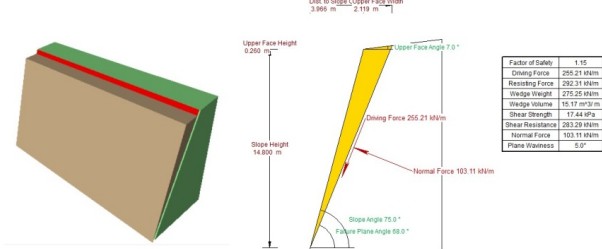

**Figure 24. Analysis of planar failure by Rocplane (slope: 75/75)**

The mechanism of wedge failure was also analyzed. Three possible wedge failure modes were identified as being the most hazardous (Fig. 25). The safety factors of the stability of wedges were calculated with the Swedge softwer (Fig. 26-28).

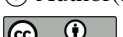



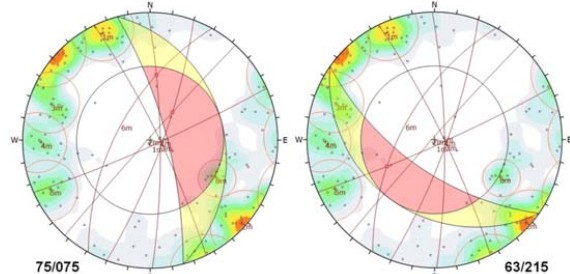

**Figure 25. Examples for the kinematic analysis of wedge failure**

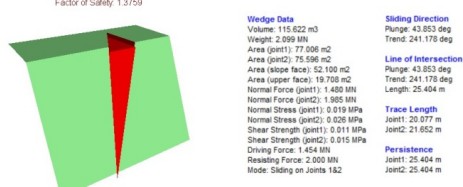

5    **Figure 26. Analysis and graphic representation of wedge failure – type 1**

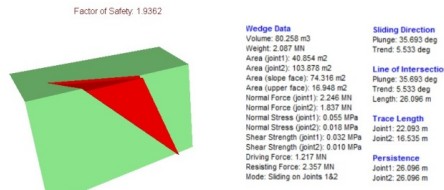

**Figure 27. Analysis and graphic representation of wedge failure – type 2**

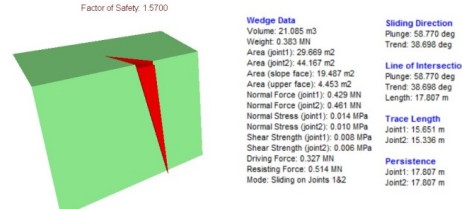

**Figure 28. Analysis and graphic representation of wedge failure – type 3**



### 4.5. Surface model resolution and engineering geological applicability of RPAS

There are three critical sets of input data in modelling of rocky slopes: i) slope geometry, ii) strength of rock mass and iii) joints system. To obtain the first – the slope geometry traditional surveying techniques were used but when the cliff is hardly accessible, these techniques do not provide appropriate data. Thus, the application of new methods such as RPAS or TLS can

help to overcome this problem. The required resolution for slope stability modelling is in the order of 10 cm, rarely 1 cm. Both techniques allow having such highly precise geometries. The drawback of these techniques is that data acquisition is a rapid process but data management and model generation take time. It has been suggested that low cost RPAS can be used in rockfall scenario analysis (Giordan et al. 2015). The combination of RPAS and TLS can also be used for fine scale modelling of erosion (Neugirg 2016). In this study, the generation of DTM and rock slope profiles were crucial for slope

stability analysis. It was not possible to obtain slope morphology by using any other method. For the determination of the strength of rock masses, we first used Schmidt hammer on site. But our field tests indicate that the application of Schmidt hammer in rock strength analysis is limited when it comes to the analysis of weak low strength rocks, such as volcanic tuff. As a consequence, laboratory analysis of samples was required to obtain reliable strength parameters. The documentation of joint system and discontinuity surfaces was first made based on field data. The measurements provided reliable data on joint

orientation; however, the joint system that was found on inaccessible cliffs was not detectable. To overcome this problem, RPAS and TSL generated images were used. The frequency of joints was observed based on these images. Similar approach was used in previous studies by Assali et al. 2014, Martino & Mazzanti 2014 and Margottini et al. 2015. It is necessary to emphasize that with the sole use of these remote sensing techniques, it is not possible to detect precise joint orientation. Consequently, field measurements are considered a must in joint system analysis.

This research has proved that the terrestrial laser scanning can achieve highly detailed and high precision point cloud about the studied surface. This point cloud is an excellent base to the elevation model, where suitable morphology analysis tools can draw the geologists' attention to the places where further examinations shall be performed.

The remotely piloted aircraft system is a relatively new platform in earth sciences (, but it has become clear that the captured imagery can be processed in a closed technology resulting in a point cloud, all of which is comparable to the TLS based one.

The resolution and quality of the obtained point cloud has been evaluated in the corresponding steps: digital elevation model and morphological analyses have been executed in the same manner as it was done with TLS data. It can be stated that the outcome of the RPAS data capture and processing technology is similarly useful for geological applications.

### 5. Conclusions

The application of terrestrial laser scanning and remotely piloted aircraft system is an excellent field data capturing technique for stability assessment of steep rock slopes. If the focus is on hazard evaluation, these technologies must be deployed very quickly, the data must be available in an easy and rapid way, and of course, the achieved information must support the required geoanalysis methods. The present study of highly dissected slope also proves that TLS is a technology where





inaccessible terrain parts can result in completeness problems. The use of RPAS technique can overcome this bottleneck: by the application of UAVs equipped with camera, relevant amount and quality imagery data can be collected about steep cliff faces. If no physical approach of a cliff is needed the field work is much safer; this technology can support even the documentation and evaluation of life-threatening locations. The vegetation-cover hides some part of the surface, but

choosing adequate data acquisition time (e.g. in canopy-free seasons), this circumstances can be managed. The collected images also provide a visual base to interpret the phenomena affecting the surface deformations. The remote sensing techniques applied provide key data on slope geometries. Without obtaining slope geometries the FEM modelling of slope is impossible. Beside slope geometries, the joint system can also be detected by TLS and RPAS. These techniques provide data mostly on joint spacing. Hence, the field documentation of joints is crucial in detecting joint orientation. In addition to those

drawbacks, at dissected and shaded areas where data gathering is not reliable, completeness problems may arise. Laboratory analyses of rock types are also necessary to obtain reliable geomechanical input parameters for the FEM model. Thus the slope stability analysis of steep rock slopes requires the use of these combined methods.

## Acknowledgements

The help of B. Czinder, B. Kleb, Z. Koppányi, B. Molnár, B. Pálinkás and B. Vásárhelyi is acknowledged. The support of

the National Research Development and Innovation (NKFI) Fund (ref. no. K 116532) is appreciated.

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
