# Peer review of "Slope stability and rock fall assessment of volcanic tuffs using RPAS"

_Natural Hazards and Earth System Sciences, 2017_

## Referee Comment (RC1) · Anonymous Referee #1 · 19 Mar 2017

The M/S aplies new tools in association with basic analyse in rock slope stability. i recommend publication

---

## Referee Comment (RC2) · Anonymous Referee #2 · 15 Apr 2017

April 15, 2017

Dear Editor, dear Authors:

General comment: This manuscript presents an analysis of volcanic tuffs instability along the southern slope of the Sirok Castel hill (Hungary) through multiple remote sensing, field and laboratory techniques. The topic fits the scope of the special issue and might meet the interest of researchers studying landslide hazard and cultural heritage conservation. Having say that, I think that the paper is not ready for publication and needs to be improved.

Specific comments:

[Figure]

1) Even if I am not an English-native speaker, I would recommend an English edit to improve sentence structure and terminology. The text is often difficult to read. Especially, the introduction and the study area description need major rewriting for sense and flow.

2) The aim of the paper is not clearly stated. In this way, also the conclusion seems to be too general and lacking of the result of the analysis.

3) The structure of the manuscript would be improved separating the Discussion section from the Result section. In the actual form, most of the results seem to be not fully described. The authors use too many figures for the description of the results but most of them are not self-explanatory.

4) The description of the study area is too general and not clearly organized. Please improve the description and add details about localization, distribution and geometric characteristic (e.g. dimension and geometry of the blocks) of the existent rock fall deposit at the base of the southern slope of the hill (e.g. page 2, line 26). Additionally, add details about the proneness to weathering of the material forming the slope. This might be a key aspect in long-term slope stability. Consider also to discuss this aspect in the text also in relation to the result of the stability analysis. Avoid to make comparison with other rocks (page 3, line 5), simply describe it in detail.

5) The authors define the RPAS as a tool that (in this case) allow to create a surface model of the study area. In my opinion, this statement does not reflect the real contribution that RPAS bring in mapping and monitoring application and might be interpreted like a "commercial description of the system". I would suggest, to underline that RPAS are simply "innovative and user friendly" platforms that offer a new sensing perspective (previously reserved only for small scale and/or very expensive investigation; e.g. airborne Lidar), reducing the time and cost of data acquisition. This perspective, or in other words the possibility to bring the camera (or the sensor) at specific positions above/around the object and to take images with specific geometries, as well as the

high repeatability, dramatically enlarged applicability of close to mid-range digital and Sfm photogrammetry and surface monitoring in general.

6) From the manuscript, it is not clear why the authors need to use both the "RPAS" photogrammetry and the TLS survey to reconstruct the topography of the slope. Especially, they state (see section 3.4) that the use of both techniques made the result difficult to manage and a specific post-processing is required to solve the redundancy of the result. Considering that the result of RPAS photogrammetry are comparable to that obtained using the TLS surveys, I would suggest use only topographic data derived from the RPAS photogrammetry for the analysis and eventually use TLS data to locally validate the reconstructed topography. In this case, they might consider change the title in: "RPAS photogrammetry for slope stability analysis in cultural heritage site, Sirok Castel hill, Hungary".

7) The method section needs to be improved adding more details about data acquisition and processing. Moreover, the authors often refer to the software used in the analysis. This is a good starting point, but it is important to specify the used criterion/procedure/equation. Please, separate the FEM global stability analysis from kinematic analysis or change the title of the section. In section 3.3, it is not clear: i) if the images were acquired using an image acquisition flight plan with a predefined frontal and side overlaps or in manual model, ii) if camera lenses were calibrated to reduce the effect of peripheral distortion that might affect/compromise the topographic reconstruction, iii) how image alinement was completed (e.g. automatic and keypoints based or picture centers coordinate based), iv) if/how the authors account for picture scale variation due to unconstrained relative elevation (in case of manual acquisition). In section 3.4, it is not clear if and how have you processed TLS point clouds for vegetation removal. Looking at figures 10a, 11a, 14a and 15a it seems that the vegetation was not removed. This compromise the topographic reconstruction of part of the slope creating local anomalies in morphological index maps.

8) In the Abstract the authors state that "joint system data were obtained from DTM

and used as input parameters. . .". However, in section 3.7, the authors state that "main discontinuity sets were measured manually on site" and TLS and UAV (RPAS) models "had been used also to determine the most hazardous part of the hillslope for block stability analyses" since "many parts of the hillslope cannot have been measured manually". From these sentences, it is not clear how the TLS and UAV (RPAS) contributed to discontinuity measurement and how the authors process models for discontinuity extraction. Please clarify this aspect.

9) In my opinion it is not clear which is the real contribution of morphological index maps to the study. If not supported by a specific description and comparison with field data the interpretation that the author made in the result section (i.e. "All resulting morphological maps strongly express the already eroded and potentially . . .") might be only considered a speculation. The improvement of the description of the study area (see comment 4) might make easier the contextualization of these maps for the understanding of the ongoing slope evolution processes.

10) The result of the stability analysis is not clearly described. Even if the author state that the critical global factor of safety is above 1, they then indicate that "the failure occurs in the weak layer". . . In this way, it is not clear what the reader should conclude looking at the analysis. Probably they would state that the slope is stable in the modeled conditions but a perturbation might induce its failure with the formation of a slip surface that should nucleate from the weaker layer. Please clarify this aspect. Additionally, from the text it is not clear if the authors account for discontinuities in the global stability analysis.

11) The number, orientation and typology of the major discontinuity systems is not stated. The graph of figure 18 is not self-explanatory.

12) Consider to delete figures 2, 11, 12, 13, and 21. In my opinion they do not add particular value to the analysis. It is not clear which parts of the slope is shown in figures 9, 10, and 14. Please add a specific map. Indicate also the localization of the

cross sections of figure 22. From the text, is not clear the number of tested sections and the width of the slope.

13) The use of references is generally appropriate. Please, thoroughly check consistency of both citations in the text and list of references.

With the above corrections, I feel the manuscript may be reconsidered for publication.

---

## Author Comment (AC1) · 4 Jun 2017

Thank you for your very positive review on our paper and your remark on the application of these new tools. We appreciate your opinion to publish this paper.

---

## Author Comment (AC2) · 10 Jul 2017

Answers to the review of interactive comment on "Slope stability and rock fall hazard assessment of volcanic tuffs using RPAS and TLS with 2D FEM slope modelling" by Ákos Török et al. Anonymous Referee #2 Reply: 10 June 2017

Answers to the reviewer #2: Thank you very much for your very constructive comments and suggestions. We have considered all of your comments and modified the manuscript accordingly. Please find the answers to your comments below.

[Figure]

Pls also find attached file - in the SUPPLEMENT:where the asnwers are marked in red. Pls. also note that only some of the revised figures are attached to this answer as uploaded file and their numbering do not follow the numbering in the revised manuscript (i.e. Fig 1. here is Fig 14 in the revised mansucript)

Original comments - April 15, 2017

Dear Editor, Dear Authors:

General comment: This manuscript presents an analysis of volcanic tuffs instability along the southern slope of the Sirok Castel hill (Hungary) through multiple remote sensing, field and laboratory techniques. The topic fits the scope of the special issue and might meet the interest of researchers studying landslide hazard and cultural heritage conservation. Having say that, I think that the paper is not ready for publication and needs to be improved. Specific comments:

1) Even if I am not an English-native speaker, I would recommend an English edit to improve sentence structure and terminology. The text is often difficult to read. Especially, the introduction and the study area description need major rewriting for sense and flow.

Answer: The revised paper is checked by a native speaker, who corrects the text.

2) The aim of the paper is not clearly stated. In this way, also the conclusion seems to be too general and lacking of the result of the analysis.

Answer: The new version of the paper has been written after considering this review. We have reformulated our goals. Slope stability analyses are in the focus supported by remotely piloted aerial systems (RPAS) and analyses by finite element methods (FEM). The used terrestrial laser scanning (TLS) was only applied for validation purposes; the revised text is written accordingly. Several figures were removed in order to make more focused content on RPAS-based survey. The slope stability analysis was revised and additional data on the location of studied sections and on the links between the data

set obtained by RPAS and used in stability analyses was emphasized. The Results and Discussions were separated. The Conclusion and outlook were rewritten.

3) The structure of the manuscript would be improved separating the Discussion section from the Result section. In the actual form, most of the results seem to be not fully described. The authors use too many figures for the description of the results but most of them are not self-explanatory. Answer: The revised paper contains new structure: we accepted the suggestion of the reviewer. We tried to write more understandable Results and Discussion sections. In the Materials and Methods section we have swapped the TLS and RPAS sections, intended more focus on RPAS as applied data acquisition and less for TLS as a validation tool.

4) The description of the study area is too general and not clearly organized. Please improve the description and add details about localization, distribution and geometric characteristic (e.g. dimension and geometry of the blocks) of the existent rock fall deposit at the base of the southern slope of the hill (e.g. page 2, line 26). Additionally, add details about the proneness to weathering of the material forming the slope. This might be a key aspect in long-term slope stability. Consider also to discuss this aspect in the text also in relation to the result of the stability analysis. Avoid to make comparison with other rocks (page 3, line 5), simply describe it in detail.

Answer: The geological conditions of the study area are described in more details in the revised manuscript. The slope geometry is described in more details. The cross-sections where slope stability was calculated are shown in the revised paper. There are no rock fall deposits at the base of the southern slope. The proneness of the tuff to weathering was emphasized in the revised text, with added new data on the properties and with new references. The comparison with other rocks has been removed from this part of the text. However, it is necessary to emphasize that the studied tuff is very similar to other tuffs in terms of properties and in terms of slope stability.

5) The authors define the RPAS as a tool that (in this case) allow to create a surface

model of the study area. In my opinion, this statement does not reflect the real contribution that RPAS bring in mapping and monitoring application and might be interpreted like a "commercial description of the system". I would suggest, to underline that RPAS are simply "innovative and user friendly" platforms that offer a new sensing perspective (previously reserved only for small scale and/or very expensive investigation; e.g. airborne Lidar), reducing the time and cost of data acquisition. This perspective, or in other words the possibility to bring the camera (or the sensor) at specific positions above/around the object and to take images with specific geometries, as well as the high repeatability, dramatically enlarged applicability of close to mid-range digital and Sfm photogrammetry and surface monitoring in general.

Answer: We have used RPAS technology to capture fine details about the rock cliff even about its generally inaccessible parts. We agree with the reviewer that this technology is "innovative and user friendly" as well as "it offers a new sensing perspective" which can naturally "reduce time and cost". The acquired imagery was processed by Structure-from-Motion technology which became very common in photogrammetry nowadays. To be able to monitor terrain surfaces, some conversions and GIS modelling were necessary. One of the messages of our paper is that these platforms are suitable for similar tasks. We have reformulated the text in section 3.1 about RPAS.

6) From the manuscript, it is not clear why the authors need to use both the "RPAS" photogrammetry and the TLS survey to reconstruct the topography of the slope. Especially, they state (see section 3.4) that the use of both techniques made the result difficult to manage and a specific post-processing is required to solve the redundancy of the result. Considering that the result of RPAS photogrammetry are comparable to that obtained using the TLS surveys, I would suggest use only topographic data derived from the RPAS photogrammetry for the analysis and eventually use TLS data to locally validate the reconstructed topography. In this case, they might consider change the title in: "RPAS photogrammetry for slope stability analysis in cultural heritage site, Sirok Castel hill, Hungary".

Answer: Thanks for the valuable remark. We have reformulated the message in order to express that RPAS technology was the primary one and TLS was only used to validate the obtained surface data. The terrain was excellent to crosscheck these two technologies, this was the reason why we wanted originally to compare the methods. Following the suggested style, we changed the order of the sections, modified (decreased) the weight of TLS and have written hopefully clear statements about the data capture. We have changed also the paper's title, although we kept the original slope stability analysis and FEM modelling. We think that our pilot site (the Sirok Castle) is just an example how these two nice tools can be combined in geological practice.

7) The method section needs to be improved adding more details about data acquisition and processing. Moreover, the authors often refer to the software used in the analysis. This is a good starting point, but it is important to specify the used criterion/procedure/equation. Please, separate the FEM global stability analysis from kinematic analysis or change the title of the section. In section 3.3, it is not clear: i) if the images were acquired using an image acquisition flight plan with a predefined frontal and side overlaps or in manual model, ii) if camera lenses were calibrated to reduce the effect of peripheral distortion that might affect/compromise the topographic reconstruction, iii) how image alinement was completed (e.g. automatic and keypoints based or picture centers coordinate based), iv) if/how the authors account for picture scale variation due to unconstrained relative elevation (in case of manual acquisition). In section 3.4, it is not clear if and how have you processed TLS point clouds for vegetation removal. Looking at figures 10a, 11a, 14a and 15a it seems that the vegetation was not removed. This compromise the topographic reconstruction of part of the slope creating local anomalies in morphological index maps.

Answer: The Materials and Methods section has been improved as the reviewer suggested. We have deleted some figures about the equipment, as well as the duplication of presenting the results. Now the surface modelling based on RPAS observations is much clearer. More details (e.g. about flight control) is given about the processing of

the imagery. There was no prior camera calibration, only simultaneous camera calibration, so this information was put into the text. GPS measurements were supported the georeferencing, which is documented in the section, too. Following the reviewer's suggestion, we have removed the TLS-oriented results to underline its validation role. With the deletion of TLS illustrations, the vegetation removal question is not relevant anymore.

8) In the Abstract the authors state that "joint system data were obtained from DTM and used as input parameters. . .". However, in section 3.7, the authors state that "main discontinuity sets were measured manually on site" and TLS and UAV (RPAS) models "had been used also to determine the most hazardous part of the hillslope for block stability analyses" since "many parts of the hillslope cannot have been measured manually". From these sentences, it is not clear how the TLS and UAV (RPAS) contributed to discontinuity measurement and how the authors process models for discontinuity extraction. Please clarify this aspect.

Answer: The main data capturing technology was based on an RPAS system. To be able to validate this dataset we performed TLS measurements. Both technologies were used to derive digital terrain (exactly surface) models (DSMs). After revising the paper, the TLS-based results were deleted and only the data quality check remained. The geological field measurements (i.e. all field works) were supported by the preliminary surface modeling results, so the manual inspections were "oriented" after the RPAS results.

9) In my opinion it is not clear which is the real contribution of morphological index maps to the study. If not supported by a specific description and comparison with field data the interpretation that the author made in the result section (i.e. "All resulting morphological maps strongly express the already eroded and potentially . . .") might be only considered a speculation. The improvement of the description of the study area (see comment 4) might make easier the contextualization of these maps for the understanding of the ongoing slope evolution processes.

Answer: We have considered the reviewer's opinion and have reduced the indices. Since the catchment area figures excellently express the similarity of the RPAS and TLS measurement, they are kept as quality comparison. The topographic wetness index can suggestively demonstrate the geological situation, the RPAS-based index image was solely kept. We want to repeat our analysis a couple of years after the first data capture to check the potential of this technology to measure the volume and map the erosion. This is not part of the current paper.

10) The result of the stability analysis is not clearly described. Even if the author state that the critical global factor of safety is above 1, they then indicate that "the failure occurs in the weak layer". . . In this way, it is not clear what the reader should conclude looking at the analysis. Probably they would state that the slope is stable in the modeled conditions but a perturbation might induce its failure with the formation of a slip surface that should nucleate from the weaker layer. Please clarify this aspect. Additionally, from the text it is not clear if the authors account for discontinuities in the global stability analysis.

Answer: The slope stability analysis was modified in the revised paper. A modified figure that shows the "weak layers" in the slope stability model was added to the revised manuscript, clearly marking the calculated slip surface at the weak layers. A new figure that describes the studied sections is now part of the revised manuscript. The difference between this model and the planar failure and wedge failure were described in more details. A figure that shows the joint orientation (and DEM model) explains better these types of potential failures.

11) The number, orientation and typology of the major discontinuity systems is not stated. The graph of figure 18 is not self-explanatory.

Answer: Former Figures 18 and 19 (now new Figures) show the strereographic projection of the measured discontinuities as a lower hemisphere projection. Each point on the stereonet represents a normal vector of a discontinuity plane. Based on the

projections six main joint sets (85/156, 88/312, 79/110, 81/089, 82/064, 61/299) can be separated.

12) Consider to delete figures 2, 11, 12, 13, and 21. In my opinion they do not add particular value to the analysis. It is not clear which parts of the slope is shown in figures 9, 10, and 14. Please add a specific map. Indicate also the localization of the cross sections of figure 22. From the text, is not clear the number of tested sections and the width of the slope.

Answer: Fig. 2, Fig. 11, Fig. 12, Fig. 13 and Fig. 21 were deleted from the text. A new Figure was added to show which parts of the slope are shown in new Figures. An additional Figure describes the location of cross sections was added to make it more clear. Out of 55 tested cross-sections 5 were chosen to analyze the global stability. Figure 22 shows two examples for the results of the analyses: Section 1 and 3 (see new Fig). Local stability analyses were not constrained to specified sections. Areas of the possible failures were determined with kinematic analyses.

13) The use of references is generally appropriate. Please, thoroughly check consistency of both citations in the text and list of references. With the above corrections, I feel the manuscript may be reconsidered for publication.

Answer: We would like to thank to the anonymous reviewer for his/her valuable time spending with our manuscript. We have considered the suggestions and prepare a revised form of the paper.

Please also note the supplement to this comment:
https://www.nat-hazards-earth-syst-sci-discuss.net/nhess-2017-56/nhess-2017-56-AC2-supplement.pdf
* * *
**Fig. 1.** New Figure, Fig. 14 Location of some studied cross sections

[Figure]

**Fig. 2.** New Figure: Fig. 12. Joint system obtained by RPAS (marked on catchment area DEM) and also measured on the field. Numbers refer to major joint systems marked on DEM map and on rose diagram

**Fig. 3.** The results of the global stability analysis of the slopes (sections 3 and 4 on Fig. 14), total displacements are marked in blue to red (lithology is indicated by letters A-D, note the weak zone)

[Figure]

**Fig. 4.** The results of the global stability analysis of the slopes (sections 3 and 4 on Fig. 14), total displacements are marked in blue to red (lithology is indicated by letters A-D, note the weak zone)

---

## Author Response (AR1)

**Answers to the reviewers**

Original title:

 **"Slope stability and rock fall hazard assessment of volcanic tuffs using RPAS and TLS with 2D FEM slope modelling".**

Modified title:

**"Slope stability and rock fall hazard assessment of volcanic tuffs using RPAS with 2D FEM slope modelling".**

**Answer to the reviewer #1:**

Thank you very much for your very positive opinion on the manuscript.

**Answers to the reviewer #2:**

Thank you very much for your very constructive comments and suggestions. We have considered all of your comments and modified the manuscript accordingly. Please find **your comments in black** and our answers to your comments in red below.

April 15, 2017

Dear Editor, dear Authors:

General comment: This manuscript presents an analysis of volcanic tuffs instability along the southern slope of the Sirok Castel hill (Hungary) through multiple remote sensing, field and laboratory techniques. The topic fits the scope of the special issue and might meet the interest of researchers studying landslide hazard and cultural heritage conservation. Having say that, I think that the paper is not ready for publication and needs to be improved.

Specific comments:

1)      Even if I am not an English-native speaker, I would recommend an English edit to improve sentence structure and terminology. The text is often difficult to read. Especially, the introduction and the study area description need major rewriting for sense and flow.

Answer: The completely rewritten and revised paper was checked by a native speaker, who corrected the text.

2)      The aim of the paper is not clearly stated. In this way, also the conclusion seems to be too general and lacking of the result of the analysis.

Answer: The new version of the paper has been rewritten after considering this review; it became shorter and more focused. We have reformulated our goals. Slope stability analyses are in the focus supported by remotely piloted aerial systems (RPAS) and analyses by finite element methods (FEM). From the original 28 Figures 10 Figures were left (more concise) and some of them were redrawn. The text is more focused on RPAS-based survey and there is a strong link to slope stability analysis. A new figure (Fig. 4) shows the applied methods and these links (flow chart). The slope stability analysis was revised and additional data on the location of studied sections are given. The Results and Discussions were completely rephrased and given in separated chapters. The Conclusion was rewritten.

3)      The structure of the manuscript would be improved separating the Discussion section from the Result section. In the actual form, most of the results seem to be not fully described. The authors use too many figures for the description of the results but most of them are not self-explanatory.

Answer: The revised paper contains new structure: we accepted the suggestion of the reviewer. We clearly separated the Results and Discussion sections. In the Materials and Methods section we have swapped the TLS and RPAS sections, intended more focus on RPAS as applied data acquisition and less for TLS as a validation tool.

4)      The description of the study area is too general and not clearly organized. Please improve the description and add details about localization, distribution and geometric characteristic (e.g. dimension and geometry of the blocks) of the existent rock fall de- posit at the base of the southern slope of the hill (e.g. page 2, line 26). Additionally, add details about the proneness to weathering of the material forming the slope. This might be a key aspect in long-term slope stability. Consider also to discuss this aspect in the text also in relation to the result of the stability analysis. Avoid to make comparison with other rocks (page 3, line 5), simply describe it in detail.

Answer: The geological conditions of the study area and the slopes are described in more details in the revised manuscript. The cross-sections where slope stability was calculated are shown in the revised paper. There are no rock fall deposits at the base of the southern slope. The proneness of the tuff to weathering was emphasized in the revised text, with added new data on the properties and with new references. The comparison with other rocks has been removed from this part of the text. However, it is necessary to emphasize that the studied tuff is very similar to other tuffs in terms of properties and in terms of slope stability.

5)      The authors define the RPAS as a tool that (in this case) allow to create a surface model of the study area. In my opinion, this statement does not reflect the real contribution that RPAS bring in mapping and monitoring application and might be interpreted like a "commercial description of the system". I would suggest, to underline that RPAS are simply "innovative and user friendly" platforms that offer a new sensing perspective (previously reserved only for small scale and/or very expensive investigation; e.g. airborne Lidar), reducing the time and cost of data acquisition. This perspective, or in other words the possibility to bring the camera (or the sensor) at specific positions above/around the object and to take images with specific geometries, as well as the high repeatability, dramatically enlarged applicability of close to mid-range digital and Sfm photogrammetry and surface monitoring in general.

Answer: The entire structure of the paper was changed in terms of RPAS application. RPAS technology was used to capture fine details even of the inaccessible part of the rock cliff. A new part in the new Discussion chapter provides information on the application of RPAS in comparison with TLS and tachimetry, showing the advantages of RPAS. We agree with the reviewer that this technology is "innovative and user friendly" as well as "it offers a new sensing perspective" which can naturally "reduce time and cost". The acquired imagery was processed by Structure-from-Motion technology which became very common in photogrammetry nowadays. To be able to monitor terrain surfaces, some conversions and GIS modelling were necessary. One of the messages of our paper is that these platforms are suitable for similar tasks but RPAS is better, faster and cheaper. We have reformulated the text.

6)      From the manuscript, it is not clear why the authors need to use both the "RPAS" photogrammetry and the TLS survey to reconstruct the topography of the slope. Especially, they state (see section 3.4) that the use of both techniques made the result difficult to manage and a specific post-processing is required to solve the redundancy of the result. Considering that the result of RPAS photogrammetry are comparable to that obtained using the TLS surveys, I would suggest use only topographic data de- rived from the RPAS photogrammetry for the analysis and eventually use TLS data to locally validate the reconstructed topography. In this case, they might consider change the title in: "RPAS photogrammetry for slope stability analysis in cultural heritage site, Sirok Castel hill, Hungary".

Answer: We have reformulated the message in order to express that RPAS technology was the primary one and TLS was only used to validate the obtained surface data. The terrain was excellent to crosscheck these two technologies. Following the suggested style, we changed the order of the sections, modified (decreased) the weight of TLS and have written clearer statements about the data capture. We have changed also the paper's title, although we kept the original slope stability analysis and FEM modelling. We think that our pilot site (the Sirok Castle) is just an example how these two nice tools can be combined in geological practice.

7)      The method section needs to be improved adding more details about data acquisition and processing. Moreover, the authors often refer to the software used in the analysis. This is a good starting point, but it is important to specify the used criterion/procedure/equation. Please, separate the FEM global stability analysis from kinematic analysis or change the title of the section. In section 3.3, it is not clear: i) if the images were acquired using an image acquisition flight plan with a predefined frontal and side overlaps or in manual model, ii) if camera lenses were calibrated to reduce the effect of peripheral distortion that might affect/compromise the topographic reconstruction, iii) how image alinement was completed (e.g. automatic and keypoints based or picture centers coordinate based), iv) if/how the authors account for picture scale variation due to unconstrained relative elevation (in case of manual acquisition). In section 3.4, it is not clear if and how have you processed TLS point clouds for vegetation removal. Looking at figures 10a, 11a, 14a and 15a it seems that the vegetation was not removed. This compromise the topographic reconstruction of part of the slope creating local anomalies in morphological index maps.

Answer: The Materials and Methods section has been improved as the reviewer suggested. We have deleted many figures (equipment, as well as the duplications). The presented surface modelling is based on RPAS observations. More details (e.g. about flight control) is given about the processing of the imagery. There was no prior camera calibration, only simultaneous camera calibration, so this information was put into the text. GPS measurements were supported the georeferencing, which is documented in the section, too. Following the reviewer's suggestion, we have removed the TLS-oriented results to underline its validation role. With the deletion of TLS illustrations, the vegetation removal question is not relevant anymore.

8)      In the Abstract the authors state that "joint system data were obtained from DTM and used as input parameters. . .". However, in section 3.7, the authors state that "main discontinuity sets were measured manually on site" and TLS and UAV (RPAS) models "had been used also to determine the most hazardous part of the hillslope for block stability analyses" since "many parts of the hillslope cannot have been measured manually". From these sentences, it is not clear how the TLS and UAV (RPAS) contributed to discontinuity measurement and how the authors process models for discontinuity extraction. Please clarify this aspect.

Answer: Joint system data was measured in the field and at inaccessible part the data set obtained by RPAS was also used for joint orientation. As we have clearly stated in the revised manuscript the main data capturing technology was based on an RPAS system. To be able to validate this dataset we performed TLS measurements. Both technologies were used to derive digital terrain (exactly surface) models (DSMs). After revising the paper, the TLS-based results were deleted and only the data quality check remained. The geological field measurements (i.e. all field works) were supported by the preliminary surface modelling results, so the manual inspections were "oriented" after the RPAS results.

9)      In my opinion it is not clear which is the real contribution of morphological index maps to the study. If not supported by a specific description and comparison with field data the interpretation that the author made in the result section (i.e. "All resulting morphological maps strongly express the already eroded and potentially . . .") might be only considered a speculation. The improvement of the description of the study area (see comment 4) might make easier the contextualization of these maps for the understanding of the ongoing slope evolution processes.

Answer: We have considered the reviewer's opinion and have removed that part from the manuscript. One new figure (Fig. 7) shows the catchment area and the joint orientation obtained from the model. We would like to repeat our analysis a few years later after the first data capture to check the potential use of this technology to measure the volume and map the erosion. This is not part of the current paper.

10)     The result of the stability analysis is not clearly described. Even if the author state that the critical global factor of safety is above 1, they then indicate that "the failure occurs in the weak layer". . . In this way, it is not clear what the reader should conclude looking at the analysis. Probably they would state that the slope is stable in the modeled conditions but a perturbation might induce its failure with the formation of a slip surface that should nucleate from the weaker layer. Please clarify this aspect. Additionally, from the text it is not clear if the authors account for discontinuities in the global stability analysis.

Answer: The slope stability analysis was modified in the revised paper. A modified figure that shows the "weak layers" in the slope stability model is now part of the revised manuscript (new Fig. 8), clearly marking the calculated slip surface at the weak layers. A new figure that describes the studied sections is now part of the revised manuscript (Fig. 3). The difference between this model and the planar failure and wedge failure were described in more details. The revised text and Figs (Fig. 7) explains better the joint orientation and types of potential failures (planar and wedge failure).

11)     The number, orientation and typology of the major discontinuity systems is not stated. The graph of figure 18 is not self-explanatory.

Answer: Former Figures 18 and 19 were removed and Fig 23-28 were compiled in 2 new figures showing the strereographic projection of the measured discontinuities and kinematic analyses of planar and wedge failures. Based on the projections six main joint sets (85/156, 88/312, 79/110, 81/089, 82/064, 61/299) are separated and given.

12)     Consider to delete figures 2, 11, 12, 13, and 21. In my opinion they do not add particular value to the analysis. It is not clear which parts of the slope is shown in figures 9, 10, and 14. Please add a specific map. Indicate also the localization of the cross sections of figure 22. From the text, is not clear the number of tested sections and the width of the slope.

Answer: Fig. 2, Fig. 11, Fig. 12, Fig. 13 and Fig. 21 were deleted from the text. A new Figure was added (Fig 3) to show which parts of the slope are shown in new Figures and also mark the location of cross sections more clearly.

Out of 55 tested cross-sections 5 were chosen to calculate the global stability. The new Figure 8 shows two examples for the results of the analyses: Section 1 and 3 (see new Fig 3). Local stability analyses were not constrained to specified sections. Areas of the possible failures were determined with kinematic analyses.

13)     The use of references is generally appropriate. Please, thoroughly check consistency of both citations in the text and list of references.

We have checked the references and citing in the text.

With the above corrections, I feel the manuscript may be reconsidered for publication.

Answer: We would like to thank to the anonymous reviewer for his/her valuable time spending with our manuscript and we do hope that the revision answer to the comments and suggestions of the reviewer.

[revised manuscript text omitted]

---

## Referee Report (RR1)

REVIEW

The paper investigate the benefits to generate a DTM using a RPAS for engineering geology application. In particular, to use the DTM for slope stability analysis. In my opinion, the innovative contribute of this paper is quite limited in particular on RPAS point of view. DTM generation using RPSA is already a consolidated techniques, in particular nowadays is quite frequently used the use of oblique images for environmental application. This aspect could be an innovative contribute, but authors not have mentioned it.

In order to be considered for publication, it is mandatory to strongly modify several parts in the paper.

Comments:

Page 1: line9: "Low strength rhyolite tuff forms steep, hardly accessible cliffs in NE Hungary. The slope is affected by rock falls." What is the sense of these sentences? They are very generic. Moreover, the RPAS is not used to generate the DTM!!! RPAS is used to collect images, point clouds (with ALS), that can be used to generate a DTM or DSM (with vegetation). RPAS is only a tool!

Page 1: line15: "The paper demonstrates that without RPAS…" This sentence is a little bit wrong, because it is possible to generate a digital model also with alternative method, as demonstrated in the last part of the paper.

Tachymetry is an old instrument, without EODM. Nowadays total station is used!! Please, replace tachymetry with total station at least!

Page 1 line 37-38…: this sentence is not clear.

Page 2, line 8-10: RPAS is not used to generate the hard surface, but it could help to acquire the images or point clouds.  It is important to focus the attention about the planning and the data processing. To be improved.

It is necessary to define the extension of the case study: how many hectares?

Page 3: fig 2: it could be better to include the in the part a) the single index (b), c),..) close to the single box, c) is not clearly defined.

Page4: fig3 : In the paper is many times mentioned the "risky zone for tourist", but it is not clearly where is this zone. Please, define this zone in the map. In this figure, north direction has to be included. Include a graphical scale.

Caption fig3 is wrong: it is mentioned fig4, but fig.5 is the correct one. In figure 5 models are described and not the area. In the caption is said "the areas shown" but fig 5 has the model.

Page 4 line 5: it is not clear the "engineering geology" aspect. In fig 4 is not mentioned, but only geology. What is the difference between geology and Eng. Geology? To be clarified.

It is mentioned that flowchart are explained, but  it is still very  "poor" . to be better explained.

Figure 4: some parts in the flowchart are not clear: connection between point cloud and cross section, or DEM and slope stability analysis.

Page 5 line4: it is not clear how the vegetation is removed. The authors declare that there are some part covered by vegetation. How the vegetation was considered? Alternatively, you have a DSM…Using TLS, it is possible to have directly a DTM.

Page 5 from line 8: "The Remotely…" is not clear. Maybe the flight was done on 21$^{st}$ … It is important to describe better the flight planning, the % of overlapping, etc. In other words, it is necessary to include a table with all information about the flight. One question: how is possible to guarantee the correct overlapping with a manual flight, with a limited vision of the UAV? What is the GSD? What is the relative height? How the camera calibration has been realized? Go pro is a very hard camera to be calibrated.

Remarks: why have you not considered to collect oblique images?
http://www.itc.nl/library/papers_2016/pres/nex_obl_ppt.pdf

The number of points in the cloud is drastically limited with respect TLS, why? You have 3 flight…In my opinion 12 million of points is very poor for your area.

Page 6 line 7: where do you have placed the GCP? Include a map.

Page 6 line 13: typos.

Page 6 line 14: a point cloud comparison is mentioned by cloud compare, but it is not shown and described. Figure 4 mentioned is related to another part. TO BE INTEGRATED. IT IS VERY IMPORTANT TO COMPARE TLS SOLUTION WITH SFM SOLUTION.

Pag7: line 2: Figure 7. In figure 7 is not shown the model. To be correct or include a new figure

Pag7: line 3: Figure57. In figure5 is not shown the CAD model, as mentioned here.

Page 7 line 8: you mention some details in figure 4, but they are not described. (as morphological index). It is necessary to define and to describe how you calculate these indexes. Equations are required

Page 7 line 12: where do you mention the engineering geology? Not clear.

Page 7 line 29: you have to define this area in the figure 3!

Page 7 line 35: …"from RPAS (DTM model figure 4)". In figure 4 is not shown a DTM…to be clarified.

Page7 line 2: " DTM and morphological…"where? It is not defined in the paper.

Caption figure 7: "…from DEM analysis (fig 4)…where?? How?...not defined in the paper.. RPAS dataset has not sense!

Page 10 line 5: figure 4 is mentioned but it is a mistake.

Page 10 line 14-15: sentence very poor. To be clarified. It is not clear the DTM analysis.

Page 11: it is not clear what is the site where the analysis is carried out.

**DISCUSSION SECTION has to be deeply reviewed.**

Page 12 line 7-11: no comparison between TLS and SFM solution is made. It is not possible to verify the conclusion here reported.

Page 12 line 15-18: to be defined better this declaration.

Page 12 line 19-26: to be defined how to realize that. SFM is a technique quite influenced by the shadow, in contrary to TLS system. Not very clear what you want to demonstrate. It is not very ease to reconstruct a correct DTM with SFM, if the data acquisition is not correctly made.

Page 13: this part is very critical. You have to better motivate how you built this table and why you have defined this index, with a literature support! The content is partially in opposite with the table 3. To be completely reviewed.

Page 13 line 21: typos tachmietry

Conclusion. This part needs a very hard review. IT could be interesting to have a comparison between the analysis made using SFM solution and using TLS dataset, just to compare the difference, accuracy and more.

The use of RPAS for photogrammetry is a consolidated technique, even in geology than ir order to be published, it is necessary to give a more innovative definition.

---

## Author Response (AR2)

**Answers to the reviewers**

Previous title:

**"Slope stability and rock fall hazard assessment of volcanic tuffs using RPAS with 2D FEM slope modelling".**

Modified title:

**"Slope stability and rock fall assessment of volcanic tuffs using RPAS with 2D FEM slope modelling".**

Major changes in the manuscript:

- Text was rewritten and afterwards checked by a native speaker
- One new table (Table 1 – with image processing data) and 2 new Figures (Fig. 6 a DTM model, and Fig. 7. Differences between RPAS and TLS point clouds obtained by CloudCompare) were added according to the suggestions of the referee
- On Fig. 3 and on the new Fig. 7 the zone of potential rockfall is now marked (it was requested by the referee)
- Additional references were added describing the RPAS as a tool in obtaining images and the SfM - DSM generation and the use of such terrain models in slope stability assessment

**Answer to the reviewer #1:**

Thank you very much for your very constructive comments and suggestions. We have considered all of your comments and modified the manuscript accordingly. Please find **your comments in black** and our answers to your comments in red below.

**REVIEW #1**

The paper investigate the benefits to generate a DTM using a RPAS for engineering geology application. In particular, to use the DTM for slope stability analysis. In my opinion, the innovative contribute of this paper is quite limited in particular on RPAS point of view. DTM generation using RPSA is already a consolidated techniques, in particular nowadays is quite frequently used the use of oblique images for environmental application. This aspect could be an innovative contribute, but authors not have mentioned it.

In order to be considered for publication, it is mandatory to strongly modify several parts in the paper.

Comments:

**Page 1: line9: "Low strength rhyolite tuff forms steep, hardly accessible cliffs in NE Hungary. The slope is affected by rock falls." What is the sense of these sentences? They are very generic. Moreover, the RPAS is not used to generate the DTM!!! RPAS is used to collect images, point clouds (with ALS), that can be used to generate a DTM or DSM (with vegetation). RPAS is only a tool!**

That part of the abstract was corrected. We agree that RPAS is a tool and it collects images that need to be processed afterwards. The revised text takes into account this and has been modified according to the referee suggestions.

**Page 1: line15: "The paper demonstrates that without RPAS…" This sentence is a little bit wrong, because it is possible to generate a digital model also with alternative method, as demonstrated in the last part of the paper.**

We have reformulated the sentence, as RPAS based technology is capable to obtain reliable terrain information and useful input for geologic analyses.

**Tachymetry is an old instrument, without EODM. Nowadays total station is used!! Please, replace tachymetry with total station at least!**

We agree that tachymetry is a traditional measurement tool and nowadays total stations are used. Tachymetry was erased from the abstract, since the manuscript does not focus on it. The text was corrected accordingly and "total station" was used throughout in the main text of the manuscript.

**Page 1 line 37-38…: this sentence is not clear.**

The sentence was corrected.

**Page 2, line 8-10: RPAS is not used to generate the hard surface, but it could help to acquire the images or point clouds. It is important to focus the attention about the planning and the data processing. To be improved.**

The data processing of RPAS was described in detail in the revised text and a comparison with TLS was also provided. Uncovered rock surfaces were studied in this paper, where the RPAS captured images provide appropriate data for DSM. A new figure (Fig. 6) shows the DSM.

**It is necessary to define the extension of the case study: how many hectares?**

The study area is marked on Fig. 3, and now the zone where rock falls occur are also marked in the revised figs (red dot-and-dash lines).

**Page 3: fig 2: it could be better to include the in the part a) the single index (b), c),..) close to the single box, c) is not clearly defined.**

Fig. 2 was reshaped according to the comment.

**Page4: fig3 : In the paper is many times mentioned the "risky zone for tourist", but it is not clearly where is this zone. Please, define this zone in the map. In this figure, north direction has to be included. Include a graphical scale.**

On the revised Fig. 3, the zone that is affected by rock fall is marked (as well as on the new DTM on the new Fig. 6.). The direction of North is also marked on revised Fig. 3.

**Caption fig3 is wrong: it is mentioned fig4, but fig.5 is the correct one. In figure 5 models are described and not the area. In the caption is said "the areas shown" but fig 5 has the model.**

The Fig numbering was checked and it was corrected throughout the paper.

**Page 4 line 5: it is not clear the "engineering geology" aspect. In fig 4 is not mentioned, but only geology. What is the difference between geology and Eng. Geology? To be clarified.**

Engineering geology is a subtopic within the broader field of geology. It is a form of applied geology. We explain its role in slope stability assessment.

**It is mentioned that flowchart are explained, but it is still very "poor" . to be better explained.**

The flow chart is explained in more details.

**Figure 4: some parts in the flowchart are not clear: connection between point cloud and cross section, or DEM and slope stability analysis.**

Additional parts were added in the text explaining the flow chart and the flow chart was revised – the links between the parts of flow chart were also corrected.

**Page 5 line4: it is not clear how the vegetation is removed. The authors declare that there are some part covered by vegetation. How the vegetation was considered? Alternatively, you have a DSM…Using TLS, it is possible to have directly a DTM.**

Vegetation removal was done manually. At the studied area, mostly bare rock faces were found. However, some corrections were made. A new Fig (Fig. 6) shows the obtained DSM.

**Page 5 from line 8: "The Remotely…" is not clear. Maybe the flight was done on 21$^{st}$ … It is important to describe better the flight planning, the % of overlapping, etc. In other words, it is necessary to include a table with all information about the flight. One question: how is possible to guarantee the correct overlapping with a manual flight, with a limited vision of the UAV? What is the GSD? What is the relative height? How the camera calibration has been realized? Go pro is a very hard camera to be calibrated.**

We have already indicated in the text that there was no prior flight planning. The weather was not exactly fine for the RPAS-flight, because it was somewhat windy. The skilled pilot could control the drone with the help of the FPV option, so the required image overlap could be ensured. The weather conditions explain the variety of the flying height (60-100 m relative flying height). The average GSD was therefore 3-5 cm on the ground. The applied software (Pix4D) has also a camera calibration option, thus no prior camera calibration was planned and executed. We agree with the reviewer that the calibration of GoPro camera is hard, but TLS was used for the validation. A comparative analysis of point clouds of RPAS and TLS is given on the new Fig. 7. On the other new figure, on Fig. 6, the ground control points that were used for referencing are also shown.

**Remarks: why have you not considered to collect oblique images?**
**http://www.itc.nl/library/papers_2016/pres/nex_obl_ppt.pdf**

Thank you for the comments and suggestion as well as for the suggested references on oblique images. We used that in the paper and cited some of those references. The field data was collected by using RPAS and the data was validated and cross checked by TLS. We believe that the obtained data quality proved that with the use of manual controlled RPAS it is also possible to obtain high quality data.

**The number of points in the cloud is drastically limited with respect TLS, why? You have 3 flight…In my opinion 12 million of points is very poor for your area.**

We fully agree that TLS has much more captured points than required. One reason is that the region of interest is smaller than the scanned area. The next reason is that the geological analyses do not require such a high resolution data set, thus we have decided to resample the data set. The amount of the RPAS points was mentioned wrong; since nearly 19 million points were derived by the photogrammetric data evaluation. The typo has been corrected in the text.

**Page 6 line 7: where do you have placed the GCP? Include a map.**

GCPs are illustrated in the newly inserted Figure 6 (DSM).

**Page 6 line 13: typos.**

Typos have been corrected.

**Page 6 line 14: a point cloud comparison is mentioned by cloud compare, but it is not shown and described. Figure 4 mentioned is related to another part. TO BE INTEGRATED. IT IS VERY IMPORTANT TO COMPARE TLS SOLUTION WITH SFM SOLUTION.**

CloudCompare is a well-known and widely used software tool to compare point clouds and the derived models. The original text has been extended by new text and new citations as well as a new comparative Figure (new Fig. 7) has been inserted showing the differences between RPAS and TLS point clouds. It marks clearly that the analyzed point clouds are very similar; as the highest difference is less than 0.1 m. Figure citation was corrected.

**Pag7: line 2: Figure 7. In figure 7 is not shown the model. To be correct or include a new figure**

It was corrected in the text.

**Pag7: line 3: Figure57. In figure5 is not shown the CAD model, as mentioned here.**

Referencing to Fig. 7. has been removed.

**Page 7 line 8: you mention some details in figure 4, but they are not described. (as morphological index). It is necessary to define and to describe how you calculate these indexes. Equations are required**

The DSM/DEM based analyses are built-in functions in GIS software SAGA. Some of these features are documented. The morphological index functionality is further referenced with the revised paper. Unfortunately these indices can not be described by using an equation, because most of them are coming from a sophisticated algorithm. These are usually too complex for a single equation, but the method was referenced in the text.

**Page 7 line 12: where do you mention the engineering geology? Not clear.**

Engineering geology is an applied geology that also deals with slope stability. It is a widely used expression, and an Engineering Geology journal also exists.

**Page 7 line 29: you have to define this area in the figure 3!**

The zones where rock falls occur are marked in the revised Fig. 3 and also on the new fig showing DTM (Fig. 6).

**Page 7 line 35: …"from RPAS (DTM model figure 4)". In figure 4 is not shown a DTM…to be clarified.**

It was explained more clearly in the revised manuscript and a new figure was also inserted in the text that shows DSM (Fig. 6).

**Page7 line 2: " DTM and morphological…"where? It is not defined in the paper.**

We guess that the reviewer refers to Page 9. The text was corrected and rewritten.

**Caption figure 7: "…from DEM analysis (fig 4)…where?? How?...not defined in the paper.. RPAS dataset has not sense!**

The figure caption of Fig. 7 (in the revised paper this figure is Fig. 9) was corrected.

**Page 10 line 5: figure 4 is mentioned but it is a mistake.**

The mistake was corrected. The correct reference is: Fig. 3.

**Page 10 line 14-15: sentence very poor. To be clarified. It is not clear the DTM analysis.**

That part of the paper was rewritten.

**Page 11: it is not clear what is the site where the analysis is carried out.**

The kinematic analyses were made for rock joints: planar and wedge failure was calculated. 6 major sets of rock joints were identified (see text and Fig. 11 and Fig. 12.). These joints are located in the area, which are marked on Fig. 3 as a zone that is affected by rock fall.

**DISCUSSION SECTION has to be deeply reviewed.**

**Page 12 line 7-11: no comparison between TLS and SFM solution is made. It is not possible to verify the conclusion here reported.**

A new figure is inserted in the text which provides information on the point clouds of RPAS and TLS (new Fig. 7). It is better explained now in the revised text, too.

**Page 12 line 15-18: to be defined better this declaration.**

Fig. 7 proves that the differences between RPAS based and TLS based terrain models are minor, and thus RPAS is an excellent tool to solve slope stability problems.

**Page 12 line 19-26: to be defined how to realize that. SFM is a technique quite influenced by the shadow, in contrary to TLS system. Not very clear what you want to demonstrate. It is not very ease to reconstruct a correct DTM with SFM, if the data acquisition is not correctly made.**

The flight was executed in leaf-free season; otherwise the most interesting areas were not covered by vegetation. If some vegetation was captured, they were filtered out from the results. The minor effect of vegetation is clearly seen on Fig. 6, but fortunately the zones that were interesting for slope stability analyses were bare cliffs and were not covered by vegetation.

**Page 13: this part is very critical. You have to better motivate how you built this table and why you have defined this index, with a literature support! The content is partially in opposite with the table 3. To be completely reviewed.**

The table was omitted and the differences were explained in more detail in the Discussion part of the revised text.

**Page 13 line 21: typos tachmietry**

Thank you for your remark, the typo has been corrected.

Conclusion. This part needs a very hard review. IT could be interesting to have a comparison between the analysis made using SFM solution and using TLS dataset, just to compare the difference, accuracy and more.

The conclusions were rewritten.

The use of RPAS for photogrammetry is a consolidated technique, even in geology than ir order to be published, it is necessary to give a more innovative definition.

New data sets are added in the paper with new figures and corrected figures. The text was also rewritten at parts focusing on the applicability of RPAS in cliff stability analysis and the appropriate resolution of RPAS validated by TLS.
* * *
**Review #2:**

**Answer to the reviewer #2:**

We really appreciate the comments and suggested corrections. We have considered your suggestions and modified the manuscript accordingly. Please find **your comments in black** and our answers to your comments in red below.

Dear Editor, dear Authors:

General comment: The manuscript has improved from the first version, but still has lingering issues that need to be addressed and does not contains enough details to make the analysis fully understandable.

Specific comments:

1)Several issues with sentence structure and grammar still persist. I would recommend a further English edit to improve sentence structure and terminology.

The terminology was updated and a native speaker corrected the English text.

2)The authors use (in the title and at some points in the text, e.g. page 3, line 9) the term hazard assessment to describe their analyses. Since the rock fall hazard assessment requires specific data (e.g. frequency and magnitude) and further analyses (e.g. trajectory modeling) that are not presented in the text, I would suggest remove it from the title and the text. Please be specific and use the appropriate terms in the description of the analysis.

This term 'hazard' has been removed from the title and also from the text.

3)It is not clear from the text how the model obtained with RPAS based photogrammetry was validated using TLS data and which is its accuracy. Please explain the method adding specific details.

The RPAS based photogrammetry was validated by using TLS. A detailed explanation is given in the revised text and a new figure (Fig. 7) was added showing the differences between the point clouds obtained by RPAS and TLS.

4) The authors state that "the rock mass failure was analyzed with the RocFall FEM software of the Rocscience (RS2)". For what I know the RocFall software is a trajectory simulation software and does not allow to analyze the stability of a slope. Please verify and in case modify the text.

In the revised paper we have explained more how the RocScience software was used and SRF factor is explained with a modified Fig (now Fig. 9, in the previous manuscript it was Fig. 8). No trajectories were modelled.

5)It is not clear from the text how discontinuity measurements were extracted from the RPAS based photogrammetric model. This is a key aspect to underline the potential of using the RPAS based photogrammetry in rock slope stability analysis and kinematic analysis of discontinuity sets. Please clarify.

The set of discontinuities are now described better, and on Fig. 11 and Fig. 12, the main joint sets are marked. A new figure Fig. 6 shows the DSM model and the text explains how the orientation of discontinuities was obtained by using the catchment area diagram (Fig. 9). It was also validated by field measurements and 6 main joint sets were identified (see text and Fig. 11 and Fig. 12).

6) At line 7 of page 8 (and in the flow chart of figure 4) the authors state that "Risk assessment was based on slope stability calculation" but no risk analysis is presented. Please modify the text considering the real contribution of the analysis.

The risk assessment was not performed but slope stability analyses were made. Accordingly, in agreement with the referee's comment, Fig. 4 was modified and the term 'risk assessment' was not used in the text.

7) The authors state that "six main joint sets (…) were identified with prevailing NE-SW direction" and refer the text to figure 7. From the text, it is not clear what criterion is used to show joint measurements (e.g. dip/strike or dip/dip direction); it seems that measurements in the text are indicated as dip/dip direction and in the rose diagram of figure 7 as dip/strike (it is not obvious for the reader). Moreover, for what I can understand looking at figure 7, it shows only 5 joint sets and two of that have a very similar strike (I suppose strike 312 and 299). Please clarify this aspect adding specific details in the text and in the caption of figure 7.

The text was rewritten. The rose diagram shows the orientation of major discontinuities (see Fig. 9). The set of discontinuities are now described better and on Fig. 11 and Fig. 12 (previously Fig. 9 and Fig. 10); the 6 main joint sets are marked. The orientation of the 6 main joint sets is also listed in the text and it was added that dip angle/dip directions are given.

8) In the cross sections of figure 8, the authors indicate the total displacement (see caption) with color ranging from blue to red and do not show any color scale to relate displacement values to colors. In this way, it is not clear what the reader should conclude looking at the analysis, also considering that despite the safety factor is above 1 the slope is subject to deformation. Please clarify this aspect and show the result of the stability analysis for all of the selected cross sections.

The total displacements are now given on the new Fig. 10 (previously Fig. 8) and the text was reshaped accordingly.

9) In the conclusions the authors state that "According to 2D FEM modeling the intercalating low strength layer is one where potential slip surface can develop causing larger scale mass movements, but at present it has low probability". Even if this interpretation might be consistent with a modeled spatial distribution of the safety factor (that is not shown in the specific figure), in my opinion the result of the global stability analysis is not suitable to estimate the probability of occurrence of a landslide. Please modify the sentence or remove.

The sentence was corrected and the conclusions were rewritten.

With the above corrections, I feel the manuscript may be reconsidered for publication.

[revised manuscript text omitted]